# SCALING GENERALIST DATA-ANALYTIC AGENTS

**Shuofei Qiao♠\*, Yanqiu Zhao♠\*, Zhisong Qiu♠\*, Xiaobin Wang◇, Jintian Zhang♠,
Zhao Bin♠, Ningyu Zhang♠†, Yong Jiang◇, Pengjun Xie◇, Fei Huang◇, Huajun Chen♠†**
♠Zhejiang University  ◇Alibaba Group
{shuofei,zhangningyu,huajunsir}@zju.edu.cn

## ABSTRACT

Data-analytic agents are emerging as a key catalyst for automated scientific discovery and for the vision of Innovating AI. Current approaches, however, rely heavily on prompt engineering or multi-agent scaffolds over proprietary models, while open-source models still struggle with diverse-format, large-scale data files and long-horizon, multi-step reasoning that real-world analytics demands. This paper introduces DATAMIND, a scalable data synthesis and agent training recipe designed to construct generalist data-analytic agents. DATAMIND tackles three key challenges in building open-source data-analytic agents, including insufficient data resources, improper training strategy, and unstable code-based multi-turn rollout. Concretely, DATAMIND applies 1) a fine-grained task taxonomy and a recursive easy-to-hard task composition mechanism to increase the diversity and difficulty of synthesized queries; 2) a knowledge-augmented trajectory sampling strategy followed by model-based and rule-based filtering; 3) a dynamically adjustable training objective combining both SFT and RL losses; 4) a memory-frugal and stable code-based multi-turn rollout framework. Built on DATAMIND, we curate DATAMIND-12K, a high-quality trajectory set spanning diverse domains, task categories, and data file formats for data-analytic tasks. Trained on DATAMIND-12K, our DATAMIND-14B achieves state-of-the-art with an average score of 71.16% on multiple data analysis benchmarks, outperforming the strongest proprietary baselines DeepSeek-V3.1 and GPT-5. Our DATAMIND-7B also performs best among all open-source models with a score of 68.10%. We also list some empirical insights gained from our exploratory trials in the analysis experiments, aiming to provide actionable insights about agent training for the community. We have released DATAMIND-12K and DATAMIND-7B,14B for the community[1].

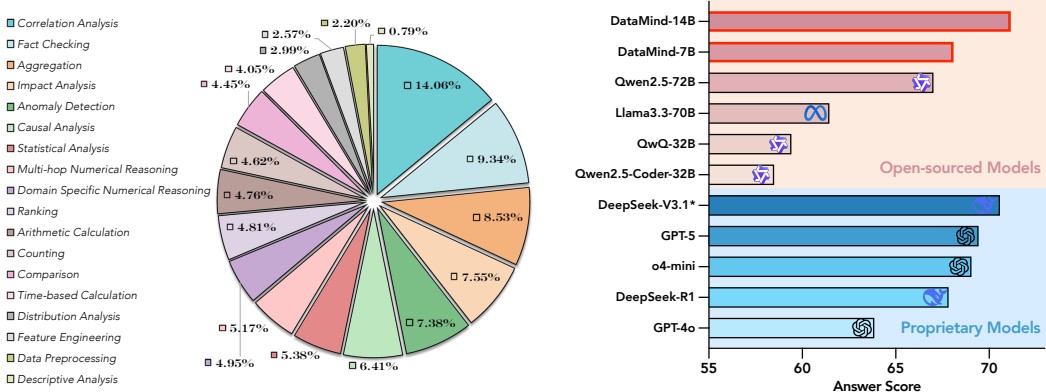

(a) Task taxonomy used in DATAMIND for fine-grained and diverse query synthesis.

(b) Performance comparison between proprietary models and open-source models on multiple datasets.

Figure 1: **(a) Task Taxonomy.** We categorize data analysis tasks into 18 fine-grained categories to enhance the diversity of our synthesized queries. **(b) Performance Comparison.** Our DATAMIND-14B achieves the best compared with all proprietary models and open-source trained or untrained models. \* Note that DeepSeek-V3.1 is an open-sourced model. We categorize it as a proprietary model for representational convenience only.

---

\*     Equal Contribution.
†     Corresponding Author.
1     Code: https://github.com/zjunlp/DataMind.

## 1 INTRODUCTION

Large Language Models (LLMs) have demonstrated formidable performance on a wide spectrum of reasoning tasks spanning math, code, and science (DeepSeek-AI et al., 2025; Kimi et al., 2025; OpenAI, 2025a; Yang et al., 2025). As AI enters its *second half* (Yao, 2025), a surge of LLM Agentic benchmarks targeted in increasingly complex and domain-specific scenarios (Jimenez et al., 2024; Starace et al., 2025; Mialon et al., 2024; Phan et al., 2025; Wei et al., 2025a) is emerging. Among them, Automated Data Analysis (Hu et al., 2024; Jing et al., 2025; Liu et al., 2024; Majumder et al., 2025), an essential pillar of AI for scientific research, plays a critical role in realizing Innovating AI and has shown its promise to boost research efficiency and accelerate scientific discovery (Chen et al., 2025b; Schmidgall et al., 2025; Lu et al., 2024; Chai et al., 2025).

Data-Analytic Agents process, model, and compute data by generating code to discover useful information or regular conclusions, thereby furnishing users with insights to support decision-making. However, existing data-analytic agents (Zhang et al., 2023; Hong et al., 2025; Li et al., 2024; Sun et al., 2025; Guo et al., 2024) are overwhelmingly built on proprietary models via prompt engineering and rely on predefined workflows or multi-agent scaffolds. The few open-source trained models (Wu et al., 2025b;c; Su et al., 2024) can only perform simple table understanding tasks (tables compact enough to fit into the prompt) and can easily break down when confronted with diverse-format, large-scale data files and long-horizon, multi-step reasoning demanded by real-world tasks.

**Challenges.** In this work, we propose to train a generalist, open-source data-analytic agent. This endeavor entails several intrinsic challenges that must be addressed: 1) *Insufficient data resources.* Training a specialized agent demands a large-scale, high-quality collection of tasks and corresponding solution trajectories. However, publicly available data analysis benchmarks often only provide a limited test set for evaluation purposes and lack step-by-step trajectory annotations, making it infeasible to assemble an effective training corpus from off-the-shelf resources. 2) *Improper training strategy.* Current agent training strategies typically follow an SFT-then-RL paradigm. Yet, in a new scenario, it remains unclear how to stabilize long-horizon agent training and how to allocate training steps across SFT and RL to achieve optimal performance. 3) *Unstable code-based multi-turn rollout.* Data files and code interpreters involve intricate memory management. Parallel agentic rollout and multi-turn code generation with limited memory resources will further exacerbate this situation.

**The DATAMIND Pipeline.** In response to the above challenges, we introduce DATAMIND, a scalable data synthesis and agent training recipe designed to build generalist data-analytic agents. To construct a large-scale training corpus, we begin by harvesting a diverse collection of data files in various formats and domains from the Internet and open communities. Then, we apply a fine-grained task taxonomy (see Fig.1a) and a recursive easy-to-hard task composition mechanism to increase the diversity and difficulty of our synthesized queries. Next, we adopt a knowledge-augmented trajectory sampling strategy to improve both the validity and reliability of synthesized trajectories. A model-based judger performs self-consistency filtering on these trajectories, followed by rule-based checks. The judgment signal will also be fed back to the model to encourage refinement, enriching the thinking patterns present in the final training set. During training, we combine SFT loss and RL loss with a dynamic coefficient to schedule the relative weight of SFT versus RL across training steps, allowing us to balance exploitation and exploration to stabilize training. For parallel multi-turn rollout, we asynchronize agent generation and code execution and utilize a chunk-wise code maintenance method to reduce peak memory usage. Moreover, we sandbox each trajectory in an isolated environment with strict caps on execution time and memory usage, enabling stable code-based multi-turn rollout.

**Results and Insights.** Through the DATAMIND pipeline, we curate **DATAMIND-12K**, a high-quality training set that spans diverse task categories and data file formats for data-analytic tasks. When trained on DATAMIND-12K, our 14B model, **DATAMIND-14B**, achieves a new state-of-the-art with an average score of 71.16% on multiple data analysis benchmarks, outperforming the strongest proprietary baselines DeepSeek-V3.1 and GPT-5 and surpassing all open-source models by a substantial margin (see Fig.1b). Our **DATAMIND-7B** also performs best among all open-source models with a score of 68.10%. Our additional analysis studies yield three valuable insights for the community: 1) Self-consistency filtering is more non-trivial than the best trajectory selection; 2) SFT loss can be an effective stabilizer for RL training, but can also be the culprit of unstable training. 3) RL can narrow the performance gap between different base models, but can hardly reverse the order.

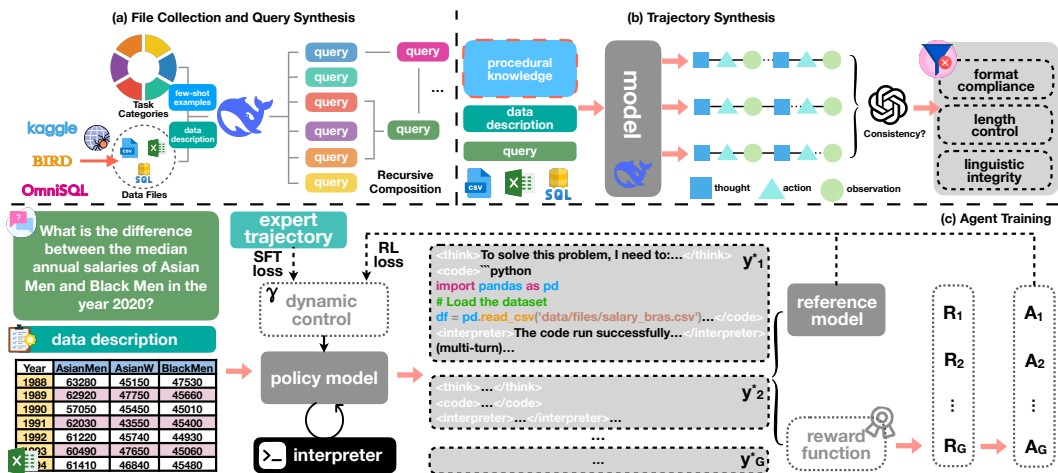

Figure 2: **The Pipeline of DATAMIND.** DATAMIND applies 1) a fine-grained task taxonomy and a recursive easy-to-hard task composition mechanism; 2) a knowledge-augmented trajectory sampling strategy followed by model-based and rule-based filtering; 3) a dynamically adjustable training objective including both SFT and RL losses; 4) a memory-frugal and stable code-based multi-turn rollout framework.

## 2  PROBLEM DEFINITION

A data analysis task $u$ is typically represented as a quadruple $u = (q, f, d, a)$, comprising the user query $q$, the data file $f$, the data description $d$, and the answer $a$, where data file $f$ may be provided in a variety of formats (.csv, .xlsx, .sqlite, etc.), and data description $d$ is optional.

Our agent framework adheres to the prevailing ReAct (Yao et al., 2023) paradigm. Upon receiving a task, the agent is required to iterate multiple rounds of Thought-Action-Observation cycles and finally produce an answer. In the data analysis scenario, Thought denotes the agent's reasoning and reflection process conditioned on the current context; Action refers to the agent's invocation of code to process and compute over the data files or the generation of the final answer. The code may be written in 🐍 Python or 🗄 SQL, depending on the data file format; Observation consists of the execution feedback returned by the environment (i.e., 🖥 Code Interpreter).

Given task $u$, let a Thought-Action-Observation loop be represented by $(\tau, \alpha, o)$, respectively. Then the agent's historical trajectory $h$ at time step $t$ can be denoted as:

$$h_t = (u, \tau_0, \alpha_0, o_0, \tau_1, \alpha_1, o_1, \ldots, \tau_{t-1}, \alpha_{t-1}, o_{t-1}). \tag{1}$$

Conditioned on the history trajectory $h_t$, the agent with parameters $\theta$ will produce its next thought $\tau_t$ and action $\alpha_t$ according to the policy $\pi_\theta(\tau_t, \alpha_t | h_t)$ and will receive an observation $o_t$ from the code interpreter after action $\alpha_t$ is executed. The whole trajectory terminates either when the agent emits an answer or when a predefined maximum number of rounds $\mathcal{T}$ is reached. **For simplicity, in the following sections**, we denote the input part provided to the agent (including $q$, $f$, and $d$) as $x$ and the trajectory (including answer $a$) sampled from the agent as $y \sim \pi_\theta(\cdot | x)$.

## 3  SCALING DATA-ANALYTIC AGENT DATA

### 3.1  FILE COLLECTION AND QUERY SYNTHESIS

**Data File Collection.**  First, we need a large amount of raw data files $f$ to scale up the potential synthesized task volume. Fortunately, the Internet and the open community benchmarks already host a massive reservoir of such files. We first target Kaggle, which contains tens of thousands of .csv and .xlsx spreadsheets. Using the official Kaggle API[2], we crawl a diverse subset of files spanning multiple domains, and then discard files that *i)* can not be loaded, *ii)* are extremely small ($< 20$ rows) or large ($> 1,000$ rows), or *iii)* contain anomalous data types. After this pipeline, we retain $3,400$ .csv and $560$ .xlsx files. For database files, we draw primarily from the training set of BIRD (Li et al., 2023b) and OmniSQL (Li et al., 2025a), both of which are high-quality corpora widely used in the Text-to-SQL field. Similarly, we sample from these sources and apply an analogous filtering pipeline, finally obtaining $1,954$ .sqlite files.

---

[2]https://www.kaggle.com/docs/api.

**Query Categorization and Synthesis.** To generate specific queries, we devise an automated script to extract meta-information $d$ of each data file, such as table headers, column names, data types, and representative rows, and then feed these metadata into DeepSeek-V3 (DeepSeek-AI, 2024) to synthesize queries $q$. To ensure both **diversity** and **fine-grainedness** of the generated questions, we refer to and refine the taxonomy in Wu et al. (2025b) and classify the data analysis tasks into 18 fine-grained categories (see Fig.1a). For each category, we carefully curate $4 \sim 6$ exemplar queries that vary in complexity and domains and serve as few-shot demonstrations. Under the guidance of these type-specific contexts, every data file is used to generate a diverse set of queries that span the full spectrum of the proposed taxonomy. To further elevate query **complexity**, we adopt a recursive easy-to-hard composition scheme that chains multiple task types, i.e., the output of one task is fed as input to the next. By iterating $2 \sim 5$ times, we progressively amplify the difficulty and create multi-hop analytic challenges that go well beyond the capability required by any single task type. The prompts for query synthesis can be found in Appx.I.2.

## 3.2 EXPERT TRAJECTORY SAMPLING AND FILTERING.

**Knowledge Augmented Trajectory Sampling.** To guarantee the quality of the synthesized trajectories, we introduce a knowledge-augmented trajectory sampling framework. Initially, for each question category, we manually craft a high-level workflow $k$ that encodes procedural knowledge and steers the model during trajectory synthesis. To further boost answer quality, we impose a self-consistency filter. We sample $\mathcal{N}$ independent trajectories per query and employ a judge model $\mathcal{M}$ powered by GPT-4o-mini (OpenAI, 2024b) to verify whether their final answers are consistent with reasoning rationales. Only trajectories that converge to the same answer are retained; among them, the judge model will also select the most concise and accurate one as our training instance $y$:

$$\{c, s, y\} = \mathcal{M}(\{y^i\}_{i=1}^{\mathcal{N}}), \quad \{y^i\}_{i=1}^{\mathcal{N}} \sim \pi_{\theta_{\text{expert}}}(\cdot|k, x), \quad y = \begin{cases} y^i \in \{y^i\}_{i=1}^{\mathcal{N}}, & s = 1 \\ \texttt{none}, & s = 0 \end{cases}, \quad (2)$$

where $c$ is the chain-of-thought process of the judge model to reach the binary conclusion $s$ of whether the sampled trajectories are consistent. We use DeepSeek-V3.1 (DeepSeek-AI, 2025) as our expert policy model $\pi_{\theta_{\text{expert}}}$. During implementation, we set $\mathcal{N} = 3$. The prompt used for trajectory sampling and the judge model $\mathcal{M}$ can be found in Appx.I.3 and Appx.I.4, respectively. We extract the final answer from the trajectory as the final synthesized answer $a$ for the corresponding query $q$. However, this pipeline inherently biases us toward easier queries whose answers are more likely to coincide. To counteract this, we refine the high-level workflow knowledge $k$ into more granular, step-by-step instructions for categories that exhibit low inter-trajectory consistency. Moreover, for trajectories that fail the consistency check, we feed the judge model's chain-of-thought back to the agent as external critique, prompting it to reflect and revise its reasoning path:

$$\{y_{\text{reflected}}^i\}_{i=1}^{\mathcal{N}} \sim \pi_{\theta_{\text{expert}}}(\cdot|k, x, \{y^i\}_{i=1}^{\mathcal{N}}, c), \quad \text{if } s = 0. \quad (3)$$

The reflected trajectories $\{y_{\text{reflected}}^i\}_{i=1}^{\mathcal{N}}$ will be fed into the judge model $\mathcal{M}$ again to conduct the consistency check and the trajectory selection in Eqn.2. This rescue loop not only salvages additional usable data but also enriches the diversity of thinking patterns embedded in the trajectory pool.

**Rule-based Trajectory Filtering.** In addition to discarding inconsistent trajectories, we apply three further rule-based filtering stages. **1) Format compliance**. We drop any trajectory that deviates from the `ReAct` format, ensuring that every remaining trajectory can be losslessly converted into our target training schema. **2) Length control**. We filter out trajectories whose final answer exceeds $1,024$ tokens, preventing the model from exploiting spurious hallucinations to artificially hit the correct string. **3) Linguistic integrity**. We remove trajectories containing garbled text or intermingled natural languages, eliminating samples that could destabilize the agent training. After the full filtering pipeline, we retain $11,707$ high-quality trajectories named as DATAMIND-12K.

## 4 SCALING DATA-ANALYTIC AGENT TRAINING

**Dynamic Control Between SFT and RL.** In this paper, we adopt a combined paradigm of Supervised Fine-Tuning (SFT) and Reinforcement Learning (RL) for the agent training. Empirically, we observe that it is difficult to strike a balance between the two stages: the model needs to absorb

sufficient knowledge from expert data during SFT, yet excessive imitation often rigidifies exploration during RL. Hence, following Zhang et al. (2025b), we employ a hybrid strategy that dynamically blends on-policy and off-policy learning, allowing the training procedure to flexibly trade off between exploitation of expert knowledge and continued exploration.

Given the training dataset $\mathcal{D}$, we express our SFT loss as:

$$\mathcal{L}_{\text{SFT}}(\theta) = -\mathbb{E}_{(x,y)\sim\mathcal{D}}\left[\sum_{t=1}^{|y|}\mathbb{I}(y_t\notin o)\cdot\log\pi_\theta(y_t|x,y_{<t})\right], \tag{4}$$

where $\mathbb{I}(y_t\notin o)$ is an indicator function that masks out any tokens produced by the environment feedback, ensuring that the model is optimized only on the agent-generated portion of the trajectory. For RL, we use the Decoupled Clip and Dynamic Sampling Policy Optimization (DAPO) (Yu et al., 2025) algorithm, minimizing the following function:

$$\mathcal{L}_{\text{DAPO}}(\theta) = -\mathbb{E}_{(x,y)\sim\mathcal{D},\{y_i^*\}_{i=1}^G\sim\pi_{\theta_{\text{old}}}(\cdot|x)}$$

$$\left[\frac{1}{\sum_{i=1}^G|y_i^*|}\sum_{i=1}^G\sum_{t=1}^{|y_i^*|}\min\left(r_{i,t}(\theta)\hat{A}_{i,t},\text{clip}\left(r_{i,t}(\theta),1-\varepsilon_{\text{low}},1+\varepsilon_{\text{high}}\right)\hat{A}_{i,t}\right)\right] \tag{5}$$

$$\text{s.t.}\quad 0 < \left|\{y_i^*\mid\texttt{is\_equivalent}(y,y_i^*)\}\right| < G,$$

where $\{y_i^*\}_{i=1}^G$ is a group of $G$ trajectories sampled from the agent policy $\pi_{\theta_{\text{old}}}$ and $y$ is the expert trajectory. Similar to SFT, any tokens emitted by the environment are discarded when computing the objective. $r_{i,t}(\theta)$ denotes the per-token importance-sampling ratio, and $\hat{A}_{i,t}$ is the advantage of the $i$-th response, obtained by normalizing the group-level rewards $\{R_i\}_{i=1}^G$:

$$r_{i,t}(\theta) = \frac{\pi_\theta(y_{i,t}^*\mid x,y_{i,<t}^*)}{\pi_{\theta_{\text{old}}}(y_{i,t}^*\mid x,y_{i,<t}^*)}, \quad \hat{A}_{i,t} = \frac{R_i - \text{mean}(\{R_i\}_{i=1}^G)}{\text{std}(\{R_i\}_{i=1}^G)}. \tag{6}$$

The inequality in Eqn.5 serves as a filtering criterion that discards trajectories lacking optimization utility whose rewards are uniformly 0 or uniformly 1 to prevent spurious gradient updates.

Finally, unlike the conventional SFT-then-RL pipeline, we jointly optimize the agent by combining the SFT and RL objectives with a dynamically balanced weighting factor:

$$\mathcal{L}_{\text{Final}}(\theta) = \gamma\mathcal{L}_{\text{SFT}}(\theta) + (1-\gamma)\mathcal{L}_{\text{DAPO}}(\theta), \tag{7}$$

where $\gamma\in[0,1]$ varies dynamically throughout training. In our implementation, $\gamma$ is initialized to a large value so that the agent first acquires knowledge from expert data via the SFT loss, and is then annealed to a small value to encourage extensive exploration through RL. Please refer to §5.4 for our analysis of different $\gamma$ settings. Importantly, for any trajectory that is filtered out by the inequality in Eqn.5, we will compute only the SFT loss. To increase the likelihood of producing eligible trajectories during the early stage of RL training, we perform a cold start using DATAMIND-12K before the process described above. We also analyze the effect of cold start for RL training in §5.4.

**Void Turns Filtering.** In multi-turn agentic training, the model can experience distributional drift due to external feedback and multi-turn compounding errors during multi-turn rollout, which will easily result in trajectory collapse, thereby destabilizing RL training (Xue et al., 2025; Baronio et al., 2025; Mai et al., 2025). We also observe this phenomenon in our experiments. To stabilize training, we directly mask out the entire loss contributed by trajectories that contain void turns. Here, a void turn is defined as an agentic loop that fails to produce a valid code snippet or answer.

**Agentic Code-based Multi-turn Rollout.** A stable environment plays a key role in stable on-policy RL training. In data-analytic agent training, massive concurrent file I/O and code execution can easily lead to environment crashes, especially with limited memory resources. To prevent this, we implement three optimizations: **1) Asynchronous interaction**. We asynchronize model generation and code execution for different data samples, which can decouple peak GPU and CPU memory demands and avoid simultaneous file I/O and code-execution spikes. **2) Chunked code maintenance**. We implement a light-weight, notebook-style code generation strategy. The model only needs to produce the code snippet required for the current reasoning step, effectively reducing generation

Table 1: **Main Results.** ♣ indicates that the original paper does not report results for the corresponding model and we use their official data and code to train the model for reproduction. ‡ denotes that we directly download their official trained model for fair evaluation. The best results for each model group are highlighted in **bold**.

| Backbone | Method | DABench | | TableBench | | BIRD | | Avg. | |
|---|---|---|---|---|---|---|---|---|---|
| | | pass@1 | pass@3 | pass@1 | pass@3 | pass@1 | pass@3 | pass@1 | pass@3 |
| *Proprietary Models* | | | | | | | | | |
| GPT-4o | | 76.39 | 84.44 | 64.97 | 75.06 | 50.20 | 62.39 | 63.85 | 73.96 |
| o4-mini | | 79.12 | 86.77 | 71.03 | 80.15 | 57.04 | 66.88 | 69.06 | 77.93 |
| DeepSeek-R1 | ReAct | 78.73 | 87.55 | 68.96 | 79.52 | 55.80 | 66.17 | 67.83 | 77.75 |
| DeepSeek-V3.1 | | **81.32** | **89.49** | **72.52** | **81.68** | 57.89 | **68.12** | **70.58** | **79.76** |
| GPT-5 | | 78.21 | 85.21 | 69.93 | 78.37 | **60.17** | 65.19 | 69.44 | 76.26 |
| *Open-source Models* | | | | | | | | | |
| Qwen-2.5-Coder-32B | | 73.15 | 81.32 | 61.11 | 72.26 | 41.20 | 60.17 | 58.49 | 71.25 |
| QwQ-32B | ReAct | 70.17 | 85.21 | 57.79 | 75.19 | 50.30 | 64.21 | 59.42 | 74.87 |
| Llama-3.3-70B | | 69.78 | 80.16 | 55.47 | 70.36 | 59.10 | 68.58 | 61.45 | 73.03 |
| Qwen-2.5-72B | | **75.33** | **86.38** | **65.44** | **76.21** | **60.30** | **69.49** | **67.02** | **77.36** |
| | ReAct | 15.05 | 35.41 | 11.70 | 28.63 | 7.02 | 18.71 | 11.26 | 27.58 |
| | TableLLM♣ | 36.71 | 71.98 | 41.01 | 70.36 | 11.99 | 16.75 | 29.90 | 53.03 |
| Qwen-2.5 | Table-R1♣ | 42.54 | 78.99 | 56.36 | 63.61 | 10.69 | 13.49 | 36.53 | 52.03 |
| Coder-7B | OmniSQL‡ | 26.46 | 36.19 | 39.95 | 50.25 | 57.11 | 66.30 | 41.17 | 50.91 |
| | SQL-R1‡ | 24.90 | 34.63 | 40.84 | 50.64 | 56.78 | 66.23 | 40.83 | 50.50 |
| | **DATAMIND** | **77.30** | **87.94** | **67.60** | **79.39** | **59.41** | **69.88** | **68.10** | **79.07** |
| | ReAct | 71.21 | 83.27 | 56.96 | 69.97 | 41.76 | 59.91 | 56.64 | 71.05 |
| | TableLLM♣ | 38.26 | 74.71 | 46.44 | 76.08 | 20.99 | 28.88 | 35.23 | 59.89 |
| Qwen-2.5 | Table-R1♣ | 45.33 | 79.38 | 50.38 | 58.91 | 11.80 | 14.08 | 35.84 | 50.79 |
| Coder-14B | OmniSQL‡ | 26.46 | 39.30 | 41.98 | 52.67 | 58.80 | 67.41 | 42.41 | 53.13 |
| | SQL-R1‡ | 27.24 | 40.47 | 41.22 | 51.02 | 58.02 | 66.62 | 42.16 | 52.70 |
| | **DATAMIND** | **80.29** | **88.72** | **70.95** | **81.81** | **62.23** | **70.21** | **71.16** | **80.25** |

latency. Furthermore, whereas conventional notebook systems maintain a global variable pool, which is memory-intensive, we retain only the textual code chunks. At runtime, we concatenate the active snippet with its predecessors, yielding the same global execution effect without the memory overhead. **3) Security Control**. To ensure secure code execution, we isolate the runtime environment for each trajectory, enforce per-trajectory limits on CPU time and peak memory, and filter any snippet containing insecure function calls before execution. Additionally, we provide an **automatic package-installation** mechanism that dynamically checks and installs uninstalled Python packages.

**Reward Design.** Our reward mainly comprises three components: **format reward** $r_{\text{format}}$, **answer reward** $r_{\text{answer}}$, and **length reward** $r_{\text{length}}$. The agent is required to enclose its reasoning process within **`<think>`**...**`</think>`** tags, place any generated data-processing code between **``** and **``**, and wrap its final answer in **`<answer>`**...**`</answer>`**. The environment's execution results will be placed between **`<interpreter>`** and **`</interpreter>`**. For the answer reward, as many answers are descriptive and thus resist rule-based verification, we adopt a *model-as-judge* powered by GPT-4o-mini (OpenAI, 2024b). We engineer a dedicated LLM evaluation prompt detailed in Appx.I.4. Both $r_{\text{format}}$ and $r_{\text{answer}}$ are binary with only $0$ and $1$. To mitigate the risk of the agent hacking the answer reward by hallucinating excessive tokens, we further impose a length-based penalty to discourage overly verbose outputs. We define the length reward and the final reward as:

$$R = \begin{cases} r_{\text{length}} \cdot r_{\text{answer}}, & r_{\text{answer}} = 1 \\ 0, & r_{\text{format}} = 1, r_{\text{answer}} = 0 \\ -0.1, & r_{\text{format}} = 0, r_{\text{answer}} = 0 \end{cases}, \quad r_{\text{length}} = \begin{cases} 1, & l \leq l_{\min} \\ \frac{l_{\max} - l}{l_{\max} - l_{\min}} \cdot 0.5 + 0.5, & l_{\min} < l \leq l_{\max} \\ 0.5, & l_{\max} < l \end{cases}$$

(8)

We incentivize correct outputs. So as long as the predicted answer exactly matches the ground truth, the model will receive a high reward ($\geq 0.5$). The specific value is length-dependent: we award a full reward if the answer length $l$ is shorter than $l_{\min}$; it decays linearly to $0.5$ when the length falls between $l_{\min}$ and $l_{\max}$; any sequence longer than $l_{\max}$ incurs a fixed length penalty of $0.5$. According to our observation, we set $l_{\min}$ and $l_{\max}$ to $256$ and $1024$ respectively during our experiments.

## 5 EXPERIMENTS

### 5.1 EXPERIMENTAL SETTINGS

**Datasets and Metrics.** We evaluate our model on three datasets related to data analysis: **DABench** (Hu et al., 2024), **TableBench** (Wu et al., 2025b), and **BIRD** (Li et al., 2023b). Our evaluation protocol aligns with our answer reward method, where a judge model powered by GPT-4o-mini (OpenAI, 2024b) is used to evaluate the correctness of the final answer. We report both pass@1 and pass@3 scores for all the methods. Please refer to Appx.D for more details.

**Models and Baselines.** We compare our models with five strong proprietary models and four outstanding open-source models (see Tab.1). In addition, we select four open-source models that have been explicitly trained for data-analysis-related tasks: **TableLLM** (Wu et al., 2025b), **Table-R1** (Wu et al., 2025c), **OmniSQL** (Li et al., 2025a), and **SQL-R1** (Ma et al., 2025). We include **Qwen-2.5-Coder-7B** and **14B** (Hui et al., 2024) as our backbone models to compare different baselines. Detailed model information and reproduction protocols for all baselines are provided in Appx.E.

**Training and Inference Setups.** We use LlamaFactory (Zheng et al., 2024) for SFT training and verl (Sheng et al., 2025) for RL training. For SFT, our learning rate is $1e-5$ with a warmup ratio of 0.1 and a cosine decay schedule. Our global batch size is set to 16. For RL, we use a learning rate of $1e-6$. The batch size is 32 with a mini batch size of 4 and the number of epochs is 1. The rollout temperature is 0.7, the top-p is 1.0, and the group size $G$ is 4. We schedule $\gamma$ via cosine decay, annealing from a peak of 0.9 to a valley of 0.05. At test time, we fix the temperature to 0.7, top-p to 0.95, and an inference batch size of 5 for all evaluations. For all the processes, the maximum number of interaction rounds $\mathcal{T}$ is set to 10. Each of the training experiments can be run on a machine with 8 80G A100 GPUs within 2 days. The detailed hyperparameter information can be seen in Appx.F.

### 5.2 MAIN RESULTS

As shown in Tab.1, our 7B model, DATAMIND-7B, achieves the best among all open-source models with an average score of 68.10%. Our 14B model, DATAMIND-14B, attains an average score of 71.16% across all tasks, surpassing all proprietary models (including the latest GPT-5 and DeepSeek-v3.1) as well as all open-source alternatives. Moreover, our DATAMIND series models demonstrate robust mastery of diverse data formats and exhibit balanced performance across all datasets. By contrast, specialized models degrade sharply when confronted with unseen data. For example, OmniSQL-7B reaches 57.11% on BIRD, yet its performance on TableBench and DABench drops steeply. Note

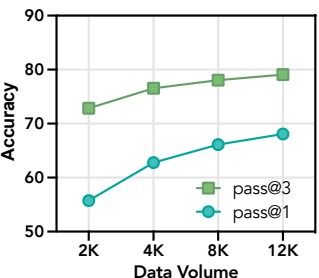

Figure 3: **Ablation on Data.**

that to ensure a fair evaluation, we have converted all tables in these two benchmarks into `.sqlite` files. Nevertheless, SQL-oriented models still underperform. This observation indicates the breadth of query types and file formats covered by DATAMIND-12K. Furthermore, TableLLM and Table-R1 are limited to small-scale tables. When evaluated on DABench's large-scale tables, they fail to generalize, and their accuracy deteriorates even further on BIRD's multi-table analysis. These results highlight our model's capacity to handle complex tabular data, which can be attributed to the difficulty distribution embedded in DATAMIND-12K. Moreover, all trained baselines are exposed to significantly larger training corpora than ours (20K instances for TableLLM and Table-R1, and 2.5M for OmniSQL and SQL-R1, versus only 12K for DATAMIND), yet we outperform them even on their adept benchmarks. This gain is attributable to the high-quality reasoning trajectories curated in DATAMIND-12K and our stable training strategy. Our model also maintains a high pass@3 score, preserving strong generation diversity while ensuring reliability.

### 5.3 ABLATION STUDIES

**Ablation on Data Volume.** We conduct an ablation study on the impact of training data scale by randomly sampling 2K, 4K, and 8K instances from DATAMIND-12K to train the agent. The average performance of the 7B model on all the benchmarks is reported in Fig.3. It shows that as the training

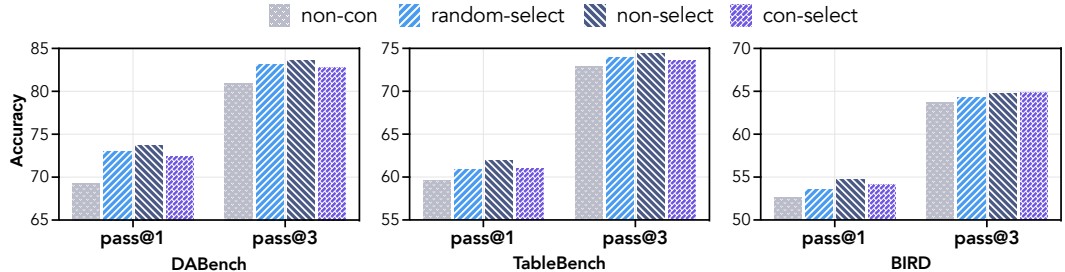

Figure 4: **Analysis on Self-Consistency Filtering and Best Trajectory Selection.** *Con-select* is our original setting, including self-consistency filtering and best trajectory selection by a judge model $\mathcal{M}$. *Non-select* uses all the sampled trajectories without the best selection. *Random-select* means randomly select a trajectory instead of the best selection. *Non-con* directly leverages all the synthesized trajectories without self-consistency filtering.

data volume increases, the model's performance exhibits a clear scaling law, demonstrating the scalability of the DATAMIND data-synthesis pipeline. Another intriguing observation is that the gap between pass@1 and pass@3 gradually narrows as the data size grows. This pattern seems to align with the conclusion in Yue et al. (2025a): RL training tends to increase the likelihood of generating correct samples, yet contributes little to expanding the model's coverage of solvable problems.

Table 2: **Ablation Results of Different Agent Training Strategy.**

| Method | Avg. | |
|---|---|---|
| | pass@1 | pass@3 |
| SFT | 62.54 | 73.74 |
| zero-RL | 58.03 | 71.72 |
| SFT-then-RL | 63.42 | 75.46 |
| **SFT-and-RL** | **68.10** | **79.07** |

**Ablation on Training Strategy.** In Tab.2, we demonstrate the superiority of our hybrid SFT+RL training objective by comparing with pure SFT, zero-RL, and SFT-then-RL on the 7B model. We need to say that the majority of our performance gains stem from our high-quality training data, as SFT alone raises the score from 11.26% (ReAct) to 62.54%, substantially closing the gap with other strong models. However, zero-RL performs markedly worse than SFT, while SFT-then-RL yields only marginal gains over pure SFT. On the contrary, our hybrid training strategy can further boost the model performance to 68.10% of pass@1 and 79.07% of pass@3. What these numbers do not reveal is the training stability conferred by our hybrid objective, as both zero-RL and SFT-then-RL can easily fall into collapse (detailed analysis about training stability can be found in §5.4), and we need to run many trials and select a relatively good checkpoint that performs best on the validation set.

**Ablation on Evaluation Method.** Because the same judge model is used during both training and evaluation, to rule out the possibility that our model wins by hacking the judge, we re-score representative baselines on DABench and TableBench with Qwen-2.5-72B serving as an external judge, and plot the pairwise mapping between the two judges in Fig.5. As shown, the rankings produced by the two judges are virtually identical (each model–dataset pair contributes two points, corresponding to pass@1 and pass@3). Quantitatively, the Pearson correlation between the two sets of scores is 0.96, indicating an almost perfect linear relationship. Furthermore, we isolate the descriptive-generation tasks in TableBench and evaluate the outputs of DATAMIND and the baselines with the dataset's original objective metric, Rouge-L, in Appx.G.4. The resulting rankings again align closely with those produced by our

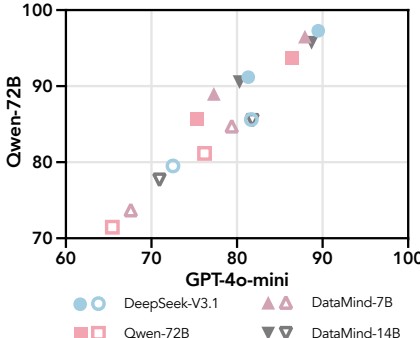

Figure 5: **Ablation on Judge Model.** Solid markers denote results on DABench; hollow markers correspond to TableBench.

*model-as-judge*. Nevertheless, Rouge-L clearly fails to capture true model performance. It over-emphasizes surface lexical and sentence overlap with the gold label rather than answer correctness. Even the powerful DeepSeek-V3.1 attains a very low Rouge-L score, underscoring the rationality and fairness of employing a judge model for evaluation.

## 5.4 ANALYSIS

**Self-consistency filtering is more non-trivial than the best trajectory selection.** In Fig.4, we analyze the impact of the self-consistency trajectory filtering and best trajectory selection strategies through SFT on the 7B model. It is evident that removing the self-consistency filtering (*non-con*)

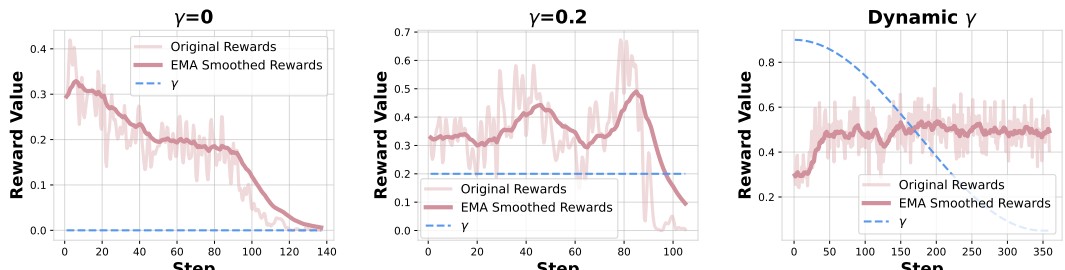

Figure 6: **The Influence of SFT Loss for RL Training.** $\gamma = 0$ denotes the absence of SFT loss, $\gamma = 0.2$ corresponds to a low SFT-loss weight, and dynamic $\gamma$ indicates our naive setting.

inflicts the most pronounced degradation on model performance: both pass@1 and pass@3 drop to varying extents across all datasets. This observation suggests that the quality of the answers produced by a trajectory is a critical guarantee of the trajectory's overall quality. Provided that the final answers are consistent, we observe that randomly selecting a single trajectory for training is not necessarily worse than explicitly choosing the best one, and it even yields a clear improvement on DABench. We hypothesize that the judge model's preference bias may potentially reduce trajectory diversity. This conjecture can be further evidenced by the pass@3 scores of *random-select*, which are on par with or superior to those of *con-select* across all three datasets. Moreover, the largest performance gains are obtained by including, without any selection, every trajectory that converges to a consistent answer. This pattern holds across all datasets and indicates that the diversity of reasoning patterns and problem-solving strategies embedded in the trajectories is more beneficial to the model's reasoning capability, which aligns with the findings in Guha et al. (2025), although we cannot fully rule out the contribution of the larger training volume introduced by this unfiltered approach.

**SFT loss is an effective stabilizer for RL training.** When our experiments are still in an exploratory phase, we use DATAMIND-12K to examine how the weight of the SFT loss in Eqn.7 influences the RL training on the 7B model without a cold start. In Fig.6, we plot the dynamics of the answer reward across training steps under different $\gamma$ settings. As can be seen, when no SFT loss is imposed ($\gamma = 0$), the answer reward declines almost monotonically. We attribute this failure to two factors. First, the 7B model's limited multi-step reasoning capability makes it difficult to roll out high-quality trajectory groups for effective learning. Second, the heterogeneity of both data structures and code languages yields highly imbalanced trajectory distributions, resulting in unstable training. Raising $\gamma$ to $0.2$ can alleviate the problem to some extent. The answer reward initially rises despite large oscillations, yet the SFT loss remains too weak to prevent the policy from eventually drifting away and collapsing. Under our dynamic-$\gamma$ schedule, the model first enjoys the stabilizing supervision of a strong SFT loss and after it matures, the SFT coefficient is gradually annealed to encourage exploration, yielding stable training during the whole process.

**SFT loss can also be the culprit of unstable training.** Although SFT loss serves as an effective stabilizer for RL training, we find that its persistent dominance throughout training can conversely trigger collapse. As shown in Fig.7, fixing $\gamma$ at a high level also causes the answer reward to rise briefly, followed by a gradual decline. The underlying reason is that over-fitting to the SFT loss traps the policy in the rigid thinking patterns embedded in the expert trajectories, especially when these trajectories are synthesized from the same model, thereby crippling exploration. To corroborate this, we track the entropy of the policy during training and observe a pronounced entropy collapse phenomenon. In contrast, our dynamic-$\gamma$ strategy can keep the policy entropy consistently at a relatively high level throughout training. Overall, we find the training process resembles raising a child. During early childhood, constant parental guidance (a large $\gamma$) is indispensable to keep the child from going astray. As the child grows up, excessive supervision stifles the child's innate drive for

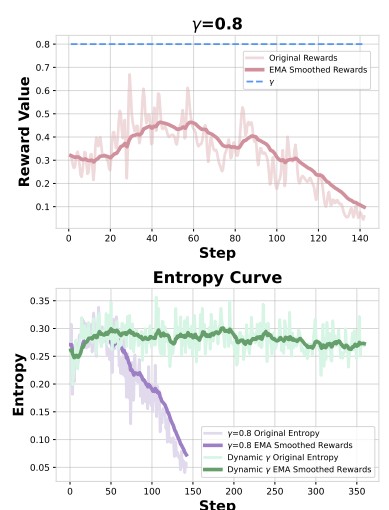

Figure 7: **Answer Reward and Entropy Dynamics** of different $\gamma$ settings.

self-directed exploration. At that stage, judiciously letting go (a small $\gamma$) enables the child to discover their true capabilities through the feedback from the surrounding world.

**RL can narrow the performance gap between different base models, but can hardly reverse the order.** Fig.8 shows the impact of different degrees of cold start on subsequent RL training. We randomly sample $3,843$ training data from DATAMIND-12K (balanced on query types) and $240$ test data (60 for each of the three test datasets) for evaluation. As the number of cold start training epochs increases, the marginal gain achieved by RL over the cold start checkpoint (i.e., the slope of the dashed line) diminishes. This indicates that RL can narrow the performance gap between different base models (Liu et al., 2025). This phenomenon also aligns well between the 7B and 14B models with the same training epochs. Nevertheless, although the gap is narrowed, post-RL performance remains positively correlated with the capability of the base model. This suggests that the bulk of knowledge is acquired during SFT, whereas RL primarily serves to unlock latent potential rather than explicitly push the model beyond its inherent capacity boundary (Yue et al., 2025a; Chu et al., 2025).

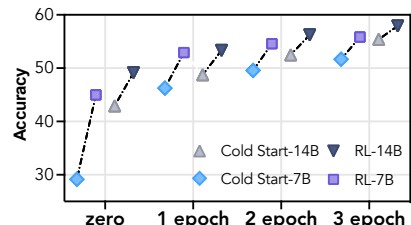

Figure 8: **Performance Gap Between Cold Start and RL** with varying cold start epochs.

## 6 RELATED WORK

**Agent Training.** The earliest wave of LLM Agents (Wang et al., 2023; Xi et al., 2023) leverages the formidable reasoning capabilities of proprietary models (Qiao et al., 2023; Chen et al., 2025a; Yao et al., 2023; Zhou et al., 2023; Hong et al., 2024; Li et al., 2023a; Wu et al., 2023). As AI entered *the second half* (Yao, 2025), numerous benchmarks targeting complex, domain-specific agentic tasks are introduced (Mialon et al., 2024; Phan et al., 2025; Jimenez et al., 2024; Chan et al., 2025; Starace et al., 2025; Wei et al., 2025a), which expose the limitations of general-purpose agent architectures, elevating domain-specific agent training to a critical necessity. The release of Large Reasoning Models (OpenAI, 2024c; DeepSeek-AI et al., 2025; Kimi et al., 2025) marks the triumph of Reinforcement Learning (RL) for LLMs. Consequently, a surge of work has sought to adapt RL algorithms to various agent domains (Jin et al., 2025a; Song et al., 2025; Li et al., 2025c; Wu et al., 2025a; Feng et al., 2025; Qian et al., 2025; Li et al., 2025b). Yet these methods presuppose a strong backbone model; researchers are therefore compelled to synthesize copious post-training data to compensate for the backbone's deficiencies. To the best of our knowledge, we are the first to systematically investigate the scaling of agent post-training in the data-analytic scenario, aiming to provide actionable insights for data synthesis and RL-driven training in other complex agent fields.

**Data-Analytic Agents and Benchmarks.** Data Analysis Agents harness the reasoning capabilities and code-generation facility of LLMs to automate the end-to-end processing of data analysis tasks. Virtually all existing data analysis agents rely on closed-source models and are limited to prompt engineering. DS-Agent (Guo et al., 2024) incorporates human insights into data analysis tasks via case-based reasoning. AutoKaggle (Li et al., 2024) decomposes the data analysis pipeline into specialized sub-tasks through a multi-agent architecture. Data-Copilot (Zhang et al., 2023) and AgenticData (Sun et al., 2025) stabilize agent behavior by orchestrating operations within predefined workflows. Data Interpreter (Hong et al., 2025) further enlarges the agent's exploration space by introducing dynamic graph-based workflows. To foster progress in this domain, numerous data analysis datasets have been introduced (Hu et al., 2024; Jing et al., 2025; Liu et al., 2024; Zhang et al., 2025a; Majumder et al., 2025). Nevertheless, each adopts its own task formulation and evaluation protocol, and the majority primarily rely on human-annotated labels. In this paper, we propose a fully automated pipeline to synthesize data analysis questions and executable code trajectories. Leveraging this synthetic corpus, we train two generalist data-analytic agents with advanced performance.

## 7 CONCLUSION

This paper introduces DATAMIND, a scalable data synthesis and agent training recipe designed to build generalist data-analytic agents. Built on DATAMIND, we curate DATAMIND-12K, a high-quality training set that spans diverse task categories and data file formats for data-analytic tasks. Trained on DATAMIND-12K, we obtain DATAMIND-7B and 14B, two advanced data-analytic agents with superior performance on multiple benchmarks compared with various proprietary and open-source baselines. We also incorporate some empirical insights gained from our exploratory trials into the analysis experiments, aiming to provide actionable insights about agentic training for the community.

ETHICS STATEMENT

This study was conducted in full compliance with established ethical standards and research best practices. All data employed are derived and synthesized exclusively from publicly available sources; no proprietary or confidential information was used. Every reference to these data sources is accurately and appropriately cited throughout the paper. We strongly encourage all users of our training dataset to uphold the highest ethical standards, ensuring fairness, transparency, and responsibility in their research. Any use of the dataset that could cause harm or negatively impact society is strictly prohibited.

REPRODUCIBILITY STATEMENT

We have submitted all our training and evaluation code in the Supplementary Material. Due to OpenReview's file size limit, we only upload a $3,843$ subset of DATAMIND-12K. We will fully open DATAMIND-12K and our models DATAMIND-7B and 14B immediately after the double blind review process. The detailed training data synthesis and agent training methods can be found in §3 and §4. We have clearly reported the details of our evaluation datasets and metrics in §5.1 and Appx.D. The detailed information about the models and baselines we use, including the model versions of proprietary models and the reproduction details of baseline models, can be found in §5.1 and Appx.E. The code framework used and the training and inference hyperparameters are mentioned in §5.1 and Appx.F. All the prompts used in our paper are presented in Appx.I.

ACKNOWLEDGMENTS

We would like to express our sincere gratitude to the anonymous reviewers for their thoughtful and constructive feedback. This work was supported by the National Natural Science Foundation of China (No. 62576307, No. NSFCU23B2055, No. NSFCU19B2027), the Fundamental Research Funds for the Central Universities (226-2023-00138), the 2025 Zhejiang Provincial Center for Disease Control and Prevention Science and Technology Talent Incubation Project (No. 2025-A-04), the 2025 Zhejiang Health Informatics Association Scientific Research Program (Key Project, No. 2025XHZN-Z01), titled "Research on Monitoring and Early Warning Methods of AI Large Model and Infectious Disease Epidemic Data Fusion", undertaken by the Zhejiang Provincial Center for Disease Control and Prevention, the Yongjiang Talent Introduction Programme (2021A-156-G), and the Information Technology Center and State Key Lab of CAD&CG, Zhejiang University.

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

## A    THE USE OF LARGE LANGUAGE MODELS

We affirm that Large Language Models are employed solely as an assisted tool to refine wording and sentence structure during our paper writing process. Their use in the experiments is strictly for scientific research purposes, and all such usage has been explicitly documented in our Experimental Settings and Reproducibility Statement. No other reliance on LLMs is involved in this work.

## B    LIMITATIONS

This paper still has some limitations that must be acknowledged: *a)* At present, we only incorporate reasoning-oriented data-analysis tasks; training, predictive, and data-visualization tasks are deliberately excluded and reserved as our important future work. *b)* Owing to computational constraints, our experimental backbone is restricted to the Qwen family, with model scale capped at 14B. Furthermore, not all mainstream benchmarks are covered in our evaluation suite. *c)* Limited by computational resources, we have not exhaustively evaluated all RL training algorithms; moreover, data scarcity constrains our RL runs to $\sim 350$ steps. In future work, we will investigate more advanced RL

strategies that enable stable, continual learning over substantially larger datasets. *d)* The current version of DATAMIND only accepts tabular data files and textual questions. In the future, we will extend DATAMIND to additional modalities.

## C  A MORE DETAILED RELATED WORK

**Agent Training.**    The earliest wave of LLM Agents (Wang et al., 2023; Xi et al., 2023) leverages the formidable reasoning capabilities of proprietary models (Qiao et al., 2023; Chen et al., 2025a). At that time, researchers primarily boost agent performance through prompt engineering (Yao et al., 2023; Zhou et al., 2023; Hong et al., 2024; Li et al., 2023a; Wu et al., 2023). To equip open-source models with agentic skills, subsequent works introduce agent training (Chen et al., 2023; Zeng et al., 2024; Chen et al., 2024; Qiao et al., 2024) via SFT. Large-scale trajectory data, manually curated or synthetically generated by closed-source models, are used to instruct-tune open-source models. As AI entered *the second half* (Yao, 2025), numerous benchmarks targeting complex, domain-specific agentic tasks are introduced (Mialon et al., 2024; Phan et al., 2025; Jimenez et al., 2024; Chan et al., 2025; Starace et al., 2025; Wei et al., 2025a), which expose the limitations of general-purpose agent architectures, elevating domain-specific agent training to a critical necessity. However, extensive studies have shown that SFT tends to drive agent models into paradigm overfitting, severely compromising their dynamic generalization ability in sophisticated agent scenarios (Chu et al., 2025; Jin et al., 2025b; Qiao et al., 2025).

The release of Large Reasoning Models (OpenAI, 2024c; DeepSeek-AI et al., 2025; Kimi et al., 2025) marks the triumph of Reinforcement Learning (RL) for LLMs. GRPO-style algorithms (Shao et al., 2024; Yu et al., 2025; Yue et al., 2025b; Wang et al., 2025a; Dong et al., 2025; Zhang et al., 2025b) enable models to autonomously explore while preserving robust generalization across diverse reasoning patterns. Consequently, a surge of work has sought to adapt GRPO-like algorithms to various agent domains (Jin et al., 2025a; Song et al., 2025; Li et al., 2025c; Wu et al., 2025a; Feng et al., 2025; Qian et al., 2025; Li et al., 2025b; Wei et al., 2025b). Yet these methods presuppose a strong backbone model; researchers are therefore compelled to synthesize copious post-training data to compensate for the backbone's deficiencies in the target agent setting. To the best of our knowledge, we are the first to systematically investigate the scaling of agent post-training data and multi-turn RL training in the data-analytic scenario, aiming to provide actionable insights for data synthesis and RL-driven training in other complex agent fields.

**Data-Analytic Agents and Benchmarks.**    Data Analysis Agents harness the reasoning capabilities and code-generation facility of LLMs to automate the end-to-end processing of data analysis tasks, constituting a critical component in the pursuit of autonomous scientific discovery (Chen et al., 2025b). Virtually all existing data analysis agents rely on closed-source models and are limited to prompt engineering. InfiAgent (Hu et al., 2024) pioneers the adoption of the `ReAct` (Yao et al., 2023) paradigm for tackling data analysis problems. DS-Agent (Guo et al., 2024) incorporates human insights into data analysis tasks via case-based reasoning. AutoKaggle (Li et al., 2024) decomposes the data analysis pipeline into specialized sub-tasks through a multi-agent architecture. Data-Copilot (Zhang et al., 2023) and AgenticData (Sun et al., 2025) stabilize agent behavior by orchestrating operations within predefined workflows. Data Interpreter (Hong et al., 2025) further enlarges the agent's exploration space by introducing dynamic graph-based workflows. To foster progress in this domain, numerous data analysis datasets have been introduced (Hu et al., 2024; Jing et al., 2025; Liu et al., 2024; Zhang et al., 2025a; Majumder et al., 2025; Wu et al., 2025b; Lei et al., 2025). Nevertheless, each adopts its own task formulation and evaluation protocol, and the majority primarily rely on human-annotated labels. In this paper, we propose a fully automated pipeline to synthesize data analysis questions and executable code trajectories. Leveraging this synthetic corpus, we train two generalist data-analytic agents with advanced performance.

## D  DATASETS AND EVALUATION DETAILS

We evaluate our model on three datasets related to data analysis. Here, we introduce the details and our evaluation protocols for each dataset:

- **DABench** (Hu et al., 2024). DABench evaluates LLMs in data analysis tasks across 257 challenges from 52 CSV files, covering 7 question categories. The original benchmark uses accuracy as the metric. The model's answer will be reformatted by an LLM to a specific structure and compared with the gold label using regular expression matching. Here, we directly utilize the *model-as-judge* to compare the predicted answer and the gold answer.

- **TableBench** (Wu et al., 2025b). TableBench is a real-world table reasoning benchmark spanning 18 fields and four major categories. The tables in TableBench are organized using `.json` files. So we first transform them into `.csv` files. Then, we filter the trend forecasting and chart generation questions because these questions do not have explicit gold answers. The original benchmark uses Rouge-L as the metric, and we apply *model-as-judge* instead.

- **BIRD** (Li et al., 2023b). BIRD is a widely used Text-to-SQL benchmark. We use it to evaluate our model's ability to analyze table-based databases. Since the BIRD test set requires official leaderboard submission, we adopt its validation set as our testbed for convenience. As SQL execution typically returns structured and often very large tables that are hard for a *model-as-judge* to assess, we instead materialize each result into a `.csv` file and perform an exact match comparison against the gold label.

We adopt accuracy as the final metric. For every model, we run three independent trials. We take the average score of the three trials as pass@1, while the union of the three trials is taken as pass@3 (i.e., success on any single trial counts as an overall success).

## E  BASELINES AND REPRODUCTION DETAILS

**Models and Baselines.**   We compare our DATAMIND with five strong proprietary models: **GPT-4o** (gpt-4o-2024-0806) (OpenAI, 2024a), **o4-mini** (o4-mini-2025-04-16) (OpenAI, 2025b), **DeepSeek-R1** (deepseek-r1-2025-0528) (DeepSeek-AI et al., 2025), **DeepSeek-V3.1** (deepseek-v3.1-nothinking) (DeepSeek-AI, 2025), **GPT-5** (gpt-5-2025-08-07) (OpenAI, 2025a). We also include four outstanding open-source models: **QwQ-32B** (Qwen, 2025), **Qwen-2.5-Coder-32B** (Hui et al., 2024), **Llama-3.3-70B** (Dubey et al., 2024), and **Qwen-2.5-72B** (Yang et al., 2024). In addition, we select four open-source models that have been explicitly trained for data-analysis-related tasks: **TableLLM** (Wu et al., 2025b) and **Table-R1** (Wu et al., 2025c) are specialized for tabular reasoning, whereas **OmniSQL** (Li et al., 2025a) and **SQL-R1** (Ma et al., 2025) are optimized for Text-to-SQL generation. We include **Qwen-2.5-Coder-7B** and **14B** (Hui et al., 2024) as backbone models to compare different baselines. We use the Instruct version for all open-source models. Here we introduce the baselines we compare with and our reproduction details:

- **REACT** (Yao et al., 2023). For untrained models, including all proprietary models and open-source models, we test them using REACT agent format in a zero-shot manner. The detailed prompt is the same with the prompt in Fig.9.

- **TableLLM** (Wu et al., 2025b). TableLLM is trained on TableInstruct (Wu et al., 2025b), the training set of TableBench, with SFT. The data volume of TableInstruct is 19,661. We directly use TableInstruct to train Qwen2.5-Coder-7B and 14B for reproduction. As TableInstruct has three prompt modes (i.e., TCoT, SCoT, PoT), we test all of them and report the best results.

- **Table-R1** (Wu et al., 2025c). Table-R1 applies a region-enhanced SFT followed by a table-aware GRPO (Shao et al., 2024) training. The training data is also from TableInstruct for both SFT and RL stages. We follow the same data split and the training pipeline of Table-R1 to train Qwen2.5-Coder-7B and 14B for reproduction. As TableInstruct has three prompt modes (i.e., TCoT, SCoT, PoT), we test all of them and report the best results.

- **OmniSQL** (Li et al., 2025a). OmniSQL is trained on a large-scale synthesized text-to-SQL dataset SynSQL-2.5M with a data volume of 2.5M. Since it already has Qwen2.5-Coder-7B and 14B version models, we directly use them for evaluation.

- **SQL-R1** (Ma et al., 2025). SQL-R1 is further trained on OmniSQL with RL. The training data for RL is sourced from a 5K subset of SynSQL-2.5M. Since it already has open-source Qwen2.5-Coder-7B and 14B version models, we directly use them for evaluation.

Note that for a fair comparison, we strictly adhere to the original input–output formats and benchmark configurations of every baseline. Specifically, for TableLLM and Table-R1, as they are both trained

Table 3: Detailed hyperparameters used in our paper.

| Stage | Hyperparameter | Value |
|---|---|---|
| Global | max number of rounds $\mathcal{T}$ | 10 |
| SFT | learning rate | 1e-5 |
| | lr scheduler type | cosine |
| | warmup ratio | 0.1 |
| | batch size | 16 |
| | training epoch | 3 |
| | cutoff length | 8192 |
| SFT+RL | learning rate | 1e-6 |
| | lr warmup style | constant |
| | lr warmup steps | 20 |
| | batch size | 32 |
| | mini batch size | 4 |
| | training epoch | 1 |
| | max prompt length | 2048 |
| | max response length | 8192 |
| | clip ratio low $\varepsilon_{\text{low}}$ | 0.2 |
| | clip ratio high $\varepsilon_{\text{high}}$ | 0.28 |
| | rollout temperature | 0.7 |
| | rollout topp | 1.0 |
| | rollout group size $G$ | 4 |
| | $\gamma$ scheduler type | cosine |
| | $\gamma$ peak value | 0.9 |
| | $\gamma$ valley value | 0.05 |
| | length reward $l_{\text{min}}$ | 256 |
| | length reward $l_{\text{max}}$ | 1024 |
| Inference | temperature | 0.7 |
| | topp | 0.95 |
| | batch size | 5 |

on TableInstruct, we evaluate them with the same prompt used in TableInstruct and converted each table file from all the benchmarks into the string representation required by their task settings before inserting it into the prompt. For OmniSQL and SQL-R1, which are both trained on Text-to-SQL tasks, we follow their native prompts and ask the models to generate complete SQL statements directly based on the table schema and the user query. In addition, we uniformly transform all benchmark data files into the `.sqlite` format via scripts to match their task requirements.

# F  TRAINING AND INFERENCE DETAILS

We use LlamaFactory (Zheng et al., 2024) for SFT training and verl (Sheng et al., 2025) for RL training. For SFT, our learning rate is $1e-5$ with a warmup ratio of $0.1$ and a cosine decay schedule. Our global batch size is set to 16. For RL, we use a learning rate of $1e-6$. The batch size is 32 with a mini batch size of 4. The rollout temperature is $0.7$, the top-p is $1.0$, and the group size $G$ is 4. We schedule $\gamma$ via cosine decay, annealing from a peak of $0.9$ to a valley of $0.05$. At test time, we fix the temperature to $0.7$, top-p to $0.95$, and an inference batch size of 5 for all evaluations. For all the processes, the maximum number of interaction rounds $\mathcal{T}$ is set to 10. Each of the training experiments can be run on a machine with 8 80G A100 GPUs within 2 days. The detailed hyperparameters employed in DATAMIND are presented in Tab.3.

# G  MORE RESULTS

## G.1  MORE BASE MODELS.

We primarily conduct experiments with the Qwen family, as it is widely acknowledged that Qwen exhibits stronger reasoning capacity and higher plasticity during RL training (Gandhi et al., 2025; Wang et al., 2025b). To verify the universality of our DataMind pipeline on different kinds of base models, we additionally train Llama-3.1-8B-Instruct (Meta, 2024) on DATAMIND-14K and the results

are reported in Tab.4. It can be observed that the DATAMIND pipeline transfers robustly across model families. Llama-8B not only surpasses Llama-70B but also performs on par with strong baselines. Its pass@3 further approaches the level of DeepSeek-V3.1 and DATAMIND-7B.

Table 4: **Results of Llama-8B Model** compared with other baselines.

| Backbone | Method | DABench | | TableBench | | BIRD | | Avg. | |
|---|---|---|---|---|---|---|---|---|---|
| | | pass@1 | pass@3 | pass@1 | pass@3 | pass@1 | pass@3 | pass@1 | pass@3 |
| DeepSeek-V3.1 | ReAct | 81.32 | 89.49 | 72.52 | 81.68 | 57.89 | 68.12 | 70.58 | 79.76 |
| GPT-5 | | 78.21 | 85.21 | 69.93 | 78.37 | 60.17 | 65.19 | 69.44 | 76.26 |
| Llama-3.3-70B | ReAct | 69.78 | 80.16 | 55.47 | 70.36 | 59.10 | 68.58 | 61.45 | 73.03 |
| Qwen-2.5-72B | | 75.33 | 86.38 | 65.44 | 76.21 | 60.30 | 69.49 | 67.02 | 77.36 |
| Qwen-2.5-Coder-7B | DATAMIND | 77.30 | 87.94 | 67.60 | 79.39 | 59.41 | 69.88 | 68.10 | 79.07 |
| **Llama-3.1-8B-Instruct** | DATAMIND | 73.67 | 87.16 | 64.49 | 76.46 | 58.52 | 69.56 | 65.56 | 77.73 |

## G.2 MORE BASELINES.

Data Interpreter has been embedded into the MetaGPT framework[3] as an off-the-shelf analytics tool. Despite our best effort to decouple its pipeline and run it in isolation, we are unable to reproduce the reported 94.93% GPT-4o score on DABench and it also cannot be adapted to SQL tasks. The same discrepancy has been independently acknowledged in another paper (You et al., 2025). Our reproduction results are shown in Tab.5. It can be found that task-specific scaffolds do not consistently outperform the vanilla ReAct paradigm, and prompts engineering for one model may fail to generalize to others. We believe that the greatest truths are the simplest. This is also why we believe **agent training**, rather than prompt engineering and scaffold constructing, is the more promising path.

Table 5: **Results of Data Interpreter** compared with other baselines.

| Backbone | Method | DABench | | TableBench | |
|---|---|---|---|---|---|
| | | pass@1 | pass@3 | pass@1 | pass@3 |
| DeepSeek-V3.1 | ReAct | 81.32 | 89.49 | 72.52 | 81.68 |
| | Data Interpreter | 67.70 | 80.93 | 61.98 | 74.52 |
| Qwen-2.5-72B | ReAct | 75.33 | 86.38 | 65.44 | 76.21 |
| | Data Interpreter | 62.26 | 75.10 | 49.90 | 58.84 |
| Qwen-2.5-Coder-7B | ReAct | 15.05 | 35.41 | 11.70 | 28.63 |
| | Data Interpreter | 54.09 | 65.37 | 27.99 | 41.82 |
| | DATAMIND | **77.30** | **87.94** | **67.60** | **79.39** |
| Qwen-2.5-Coder-14B | ReAct | 71.21 | 83.27 | 56.96 | 69.97 |
| | Data Interpreter | 60.70 | 74.71 | 40.71 | 55.82 |
| | DATAMIND | **80.29** | **88.72** | **70.95** | **81.81** |

## G.3 MORE BENCHMARKS.

Table 6: **Results on QRData.**

| Backbone | Method | QRData | |
|---|---|---|---|
| | | pass@1 | pass@3 |
| DeepSeek-V3.1 | ReAct | 60.75 | 75.67 |
| Qwen-2.5-72B | ReAct | 60.50 | 72.75 |
| Qwen-2.5-7B | DATAMIND | 57.66 | 69.34 |
| Qwen-2.5-14B | DATAMIND | **62.04** | **77.62** |

To demonstrate that DATAMIND can generalize to more challenging data-analysis benchmarks, we further evaluate on QRData (Liu et al., 2024), a dataset that features intricate causal-reasoning tasks and relies heavily on domain commonsense knowledge. We compare our DATAMIND-7B and DATAMIND-14B agents against the strongest open-source and closed-source models; results are reported in Tab.6. Our 14B model still achieves the best performance, benefiting from the comprehensive coverage of data-analysis types and domains in DATAMIND-12K, as well as the consistent performance gains brought by our stable agent training strategy. Our 7B model also delivers competitive results. However, constrained by its parameter count, it lacks the domain-specific knowledge required, which prevents it from achieving equally impressive performance.

---

[3]https://github.com/FoundationAgents/MetaGPT.

## G.4 MORE EVALUATION METRICS.

Table 7: **Rouge-L Results** of various methods compared with our *model-as-judge*.

| Backbone | Method | TableBench | |
|---|---|---|---|
| | | Rouge-L | pass@1 |
| DeepSeek-V3.1 | ReAct | 19.64 | 72.52 |
| Qwen-2.5-72B | | 17.93 | 65.44 |
| | TableLLM | 11.33 | 9.81 |
| Qwen-2.5-7B | Table-R1 | 13.86 | 25.49 |
| | **DATAMIND** | 23.41 | 76.47 |

To further validate the soundness of our *model-as-judge* protocol, we re-score the descriptive tasks in TableBench with Rouge-L and contrast these scores with the ratings previously assigned by GPT-4o-mini. The comparison is reported in Tab.7. The resulting rankings align closely with those produced by our *model-as-judge*. Nevertheless, Rouge-L clearly fails to capture true model performance. It over-emphasizes surface lexical and sentence overlap with the gold label rather than answer correctness. This renders the performance differences among models marginal. Even the powerful DeepSeek-V3.1 attains a very low Rouge-L score, underscoring the rationality and fairness of employing a judge model for evaluation.

## H DATA CONTAMINATION ANALYSIS

Table 8: **Table Header Overlap Ratio** between DATAMIND and all the benchmarks.

| | DABench | TableBench | BIRD |
|---|---|---|---|
| **DATAMIND** | 1.29% | 0.00% | 0.02% |

To the best of our knowledge, the DABench data files are harvested from GitHub, whereas the tables in TableBench are extracted from Wikipedia, with neither overlapping with the corpora we curate. We sample the training split of BIRD and adopt synthetically generated schemas provided by OmniSQL. Consequently, our sources are also disjoint from the BIRD test set. To quantify this claim, we compute the header-overlap ratio between every table in DATAMIND-12K and the headers of each benchmark in Tab.8. The resulting overlap is exactly 0% (or statistically indistinguishable from 0%) for every benchmark, demonstrating that DATAMIND-12K introduces no data contamination into the evaluation.

## I PROMPTS USED IN OUR PAPER

### I.1 TRAINING AND EVALUATION PROMPT

---
**Training and Evaluation Prompt**

You are an expert-level data analyst and statistician who solves any data challenge through rigorous logic, systematic planning, and deep investigation. Your primary task is to answer user questions by analyzing the provided data source. You can solve the given problem step by step by utilizing Python code execution (for CSV files) or SQL queries (for database files) to support your reasoning.

**# Problem-Solving Protocol**
1. You should think through the problem logically, outlining your reasoning process in <think> and </think> tags.
2. After reasoning, write the appropriate code to execute your plan. Place your code between  and  tags.
- For CSV files and Excel files, write Python code using libraries like pandas, numpy, sklearn, etc. to analyze the data. The format should be:
```python
<your python code here>
```
- For database files, you should only use the 'get_db_info' function to view the database structure information and 'execute_sql' function to execute sql queries, and save the results. Example:
```python
get_db_info()
```
---

```
```
or
```python
execute_sql(sql="a valid SQL query here", output_path="result.csv")
```
```

3. The execution results will be returned in <interpreter> and </interpreter> tags.
4. Every time you get the code execution result, you must conduct reasoning and analyze these results carefully between <think> and </think>. If the result indicates an error or unexpected behavior, explain the issue and rewrite the previous step code. If the result indicates the code ran successfully, analyze whether the original problem has been fully solved.
- If it has been solved, explain your reasoning and then provide the final answer wrapped in <answer>...</answer>.
- If not, continue reasoning and provide the next step of code based on your previous correct code.
4. Whenever you're confident about the result, you can directly provide your answer to the question inside <answer>...</answer>.
- For CSV files and excel files, you should directly provide your answer. Make it concise and to the point. (e.g. <answer>The final answer is 3, ...</answer>)
- For database files, you must tell me the file name of the result CSV file. (e.g. <answer>The final answer is saved in the CSV file named 'result.csv'.</answer>)

# CSV File and Excel File Analysis Notes
1. Load data from 'data/files' directory using the specified CSV or Excel filename. Temporary files can be saved to 'data/tmp'.
2. In your first step, you should use print() to inspect the data columns, the first 3 rows, and the type of the columns and so on to understand the data structure.
3. If you want to get the value of a variable in your code, you must print it out using print() (e.g., print(f"max: {{max}}")) to understand the current value and state of variables.
4. Only proceed to the next step of code if the current step is written correctly. Each step must build on the previous code.

# Database File Analysis Notes
1. You can only use sqlite database engine to execute your SQL queries.
2. In your first step, use get_db_info() to inspect the database schema.
3. In your answer, you must provide the file name of the result CSV file. Make sure the answer file has been saved in the current directory.

# Additional Notes:
1. Avoid including irrelevant commentary outside of the designated tags <think>, , <interpreter>, and <answer>.
2. If the last step is not correct, you should first conduct a deep analysis of the previous step and then rewrite the code to fix the issue.
3. Keep your responses concise, structured, and directly tied to the original question.

# Data Source
**The data source path is '{data_source_path}'.**

Figure 9: Prompt for Training and Evaluation.

## I.2 QUERY SYNTHESIS PROMPT

---

**Query Synthesis Prompt**

You are a data analysis expert and assistant. Your task is to generate high-quality, insightful data analysis questions based on a given data file's metadata, headers, and column descriptions. You always consider the semantics of the data and produce questions that can support exploratory data analysis, business understanding, or hypothesis generation. Your output should be clear, structured, and directly usable for data analysis planning.

**# Objective**
Based on the information from the data file and the specified question type, generate one unique and meaningful exploratory data analysis (EDA) question. The phrasing and structure of the question must closely follow the style of the provided examples below. The goal is to generate questions that can reveal various types of insights from the dataset.

—

**## Here is some meta information about the data file:**

### Description:
{description.replace('#', '###')}

### Header:
{meta_info['head']}

### Columns:
{meta_info['columns']}

### Columns Type:
{meta_info['type']}

### Columns Range:
{meta_info['range']}

### Columns Unique Values:
{meta_info['unique']}

### Row Number:
{meta_info['row_count']}

### Column Number
{meta_info['column_count']}

—

**## EDA Question Types with Short Descriptions**
1. **Aggregation** - Summarize data values to understand overall patterns or trends, such as calculating averages, totals, or maximums. Often used to quantify general behavior across a dataset.
2. **Ranking** - Identify and compare items based on specific metrics to determine their relative standing, often highlighting the highest, lowest.
3. **Counting** - Determine the number of items that meet specific conditions or criteria. This typically involves filtering data and counting matching entries.
4. **Comparison** - Compare values across data points to identify differences, similarities, or extremes, often focusing on identifying the highest, lowest, or range between values.
5. **Domain Specific** - Analyze data within a specific field or context using domain knowledge to interpret results, answer specialized questions, or derive insights meaningful to

that area.

6. **Causal Analysis** - Analyzing relationships beyond correlation, often requiring controlled experiments or advanced statistical methods to infer causality.

7. **Statistical Analysis** - Apply statistical measures (e.g., median, standard deviation, variance, growth rate) to summarize, describe, or evaluate patterns and variability within the data.

8. **Correlation Analysis** - Measure the strength and direction of the relationship between two quantitative variables, typically using a correlation coefficient to assess how closely the variables move together.

9. **Arithmetic Calculation** - Perform basic mathematical operations (e.g., addition, subtraction, multiplication, division) to compute totals, differences, or projections based on the given data.

10. **Descriptive Analysis** - Provide an overview of the dataset by explaining its structure, key columns, and any observable patterns or trends. Focus on summarizing what the data shows without drawing causal inferences.

11. **Impact Analysis** - Analyze relationships between variables to determine how one factor influences another. This involves identifying trends, correlations, or causations within the data to assess impact over time or across categories.

12. **Fact Checking** - Retrieve and verify multiple related facts across different data points to answer a question. This requires connecting and cross-referencing information from various parts of a dataset.

13. **Anomaly Detection** - Identify data points that significantly differ from the expected pattern or norm, potentially indicating errors, outliers, or unusual behavior.

14. **Multi-hop Numerical Reasoning** - Perform numerical reasoning that requires combining multiple pieces of information or steps. This often involves intermediate calculations or logical sequencing to reach the final answer.

15. **Time-based Calculation** - Analyze data across time periods to identify trends, changes, or cumulative values, often involving comparisons between different time intervals or calculating growth rates over time.

16. **Distribution Analysis** - Analyze how data values are distributed for a given variable or across groups. This includes assessing normality (e.g., via the Shapiro-Wilk test), skewness, kurtosis, or comparing distributions between groups using statistical tests (e.g., the Mann-Whitney U). It helps in understanding the shape, spread, and symmetry of the data, and whether it meets assumptions required for other analyses.

17. **Feature Engineering** - Create, transform, combine, or extract variables to enhance data quality or modeling potential. This includes generating new columns, deriving ratios or indicators, aggregating related values, or reformatting data to reveal deeper patterns or prepare for predictive analysis.

18. **Comprehensive Data Preprocessing** - Perform a sequence of data cleaning and preparation steps to ensure the dataset is ready for analysis or modeling. This includes handling missing values, transforming data types, encoding categorical variables, normalizing or scaling numerical features, and correcting inconsistencies. The goal is to produce a clean, well-structured, and analysis-ready dataset.

—

## Analysis Question Type Focus

### Primary Question Type:
The question should primarily focus on the given type:
{question_info['question_type']}

—

## Requirements:
- Return exactly one data analysis question per execution.
- The question should mainly focus on the specified question type:

{question_info['question_type']}} - Use clear, concise, and practical language suitable for data analysts.
- The question should be grounded in the context and structure of the data file.
- **The phrasing and style of the question must closely mirror the examples provided**. This includes using similar formats, tones, and syntactic patterns.
- **Follow the same linguistic conventions as the examples**. For example, if the examples use formulations like "What is the difference between...", "How many...", or "Which of the following...", then your generated question must follow similar patterns to ensure stylistic consistency.
- **Output format must include**:
- The question inside '<question>...</question>'
- A short description inside '<description>...</description>'
- The question type inside '<type>...</type>'

—

## Template Enforcement
You are strictly required to follow the templates provided for the question type below:
{get_question_template(question_info['question_template'])}

- You must use the template to generate the corresponding question.
- Generate the question by using the template with relevant columns, values, or descriptions from the dataset.
- Do not introduce new question styles, structures, or alternative phrasings.

This is a hard constraint, not a suggestion.

—

## Here are some examples:
{question_info['question_example']}

—
## Now, generate a high-quality EDA question:

Figure 10: Prompt for Query Synthesis.

### I.3  TRAJECTORY SAMPLING PROMPT

Trajectory Sampling Prompt

You are a Data Analysis Assistant who can solve the given problem step by step with utilizing a code execution tool to support your reasoning.

1. You should think through the data analysis problem logically, outlining your reasoning process in the <reasoning>...</reasoning> tags.

2. After reasoning, you can write Python code if necessary and use print() statements to inspect key values to support your reasoning. Remember to place your Python code inside ```python ... ``` tags. You may use libraries like pandas, numpy, sklearn, etc.

3. User will execute the code and return results in <interpreter>...</interpreter>. Every time you get the code execution result, you must conduct reasoning and analyze these results carefully between <reasoning> and </reasoning>, following this process:
Whether the result indicates an error or unexpected behavior, explain the issue and rewrite

the previous step code, or the result indicates the code ran successfully, analyze whether the original problem has been fully solved.
- If it has been solved, explain your reasoning and then provide the final answer wrapped in <answer>...</answer>.
- If not, continue reasoning and provide the next step of code based on your previous correct code.

4. Whenever you're confident about the result, you can directly provide your answer to the question inside <answer>...</answer>.

5. Format Example:
- Code Format:
<reasoning>
Your reasoning here, step by step, explaining your thought process and how you will approach the problem.
</reasoning>


```python
Your Python code here, using print() statements to inspect variables.
```


- Answer Format:
<reasoning>
Your final reasoning here. Summarize the solution to the question.
</reasoning>

<answer>
Your final answer to the question, keep your answer as brief as possible while ensuring it is complete, concise, and clearly stated. **No longer than 1024 words.**
</answer>

6. Important Notes:
- To read data files, load from the '{file_dir}' using the file '{question['filename']}'.
- **If you want to get the value of a variable in your code, you must print it out using print() (e.g., print(f"max: {{max}}"))** to understand the current value and state of variables. Try to use one or two print() statements in one single step to avoid overwhelming the user with too many outputs.
- Don't use visualization libraries like matplotlib or seaborn, as the user will not be able to see the plots.
- Only proceed to the next step of code if the current step is written correctly. Each step must build on the previous code.
- Avoid including irrelevant commentary outside of the designated tags <reasoning>, , <interpreter>, and <answer>.
- Keep your responses concise, structured, and directly tied to the original question.

7. When beginning to solve a problem, you are encouraged to follow these two initial steps as a general guideline (not a rigid rule):
- Step 1: Load the Excel file specified in the question using pandas.read_excel() from the path '{file_dir}/{question["filename"]}'. Then, inspect the dataset structure by printing:
- the first 3 rows using df.head(3)
- the list of column names using df.columns
- optionally, the data types using df.dtypes if helpful for understanding column types.
- Step 2: Analyze the question to determine which columns or rows in the dataset are relevant. Then apply appropriate operations to those columns/rows to address the problem.
- For complex tasks, you may adapt or break the second step into smaller sub-tasks and solve

> them incrementally. This approach helps manage complexity and ensures accurate reasoning throughout the analysis.
> - Error Handling: If your code contains a bug or raises an exception during execution, first focus on debugging and resolving the issue before proceeding with solving the main problem.
>
> {get_few_shot(question['question_type'])}

Figure 11: Prompt for Trajectory Sampling.

## I.4 JUDGE MODEL PROMPT

> **Judge Model Prompt (Self-consistency)**
>
> You are a precision evaluation system. Your task is to determine whether three AI-generated answers are equivalent and fully correct. Use the following evaluation criteria:
>
> 1. Semantic Equivalence: For descriptive questions, all answers must express the same meaning, even if phrased differently.
> 2. Numerical Agreement: For numerical/calculation questions, any numeric values must be within 3% of each other. Convert all units to be comparable before comparing. If a number is derived through a formula, check that the process is logically sound.
> 3. Completeness: The final answer must fully address all aspects of the original question. Do not ignore implicit sub-questions.
> 4. Source Traceability: Identify which answer (1, 2, or 3) most clearly and accurately represents the final synthesized answer.
>
> Respond using:
> - <reasoning>...</reasoning>. Please use this tag to think through the evaluation process step by step, explaining your reasoning and how you will choose the best answer. Be careful to consider all three answers.
> - <correct>...</correct>. Use 'yes' or 'no' to indicate whether all three answers are semantically equivalent and fully correct. It's fine if the differences are justifiable.
> - <number>...</number>. Use 1, 2, or 3 to choose the single best answer that should be used as the final response. If you cannot determine a clear best answer, choose the one that is most complete and accurate.
>
> ### Question
> question
>
> ### Answer 1
> answer_1
>
> ### Answer 2
> answer_2
>
> ### Answer 3
> answer_3
>
> Evaluate standard:
> 1. Are all three answers semantically equivalent and/or numerically consistent ($\leq 3\%$ difference)?
> 2. Do all three answers fully resolve the original question without omissions?
> 3. Which single answer is the best basis for a final polished response?
>
> Respond in this format:
> <reasoning>...</reasoning>

<correct>yes</correct> or <correct>no</correct>
<number>...</number>

Figure 12: Prompt for Consistency Judge Model.

**Judge Model Prompt (Reward)**

You are a fair and professional evaluator. Your task is to assess how closely an AI assistant's answer matches the provided ground truth for a given question. You are to provide a numerical score for how well the response answers the question based on the ground truth answer.

Your evaluation should focus on the assistant's answer to the question. Begin your evaluation by comparing the assistant's answer with the ground_truth answer. Identify and correct any mistakes. Be as objective as possible.

Evaluate the correctness (0 for incorrect, 1 for correct) of the predicted answer to the question:

Question: {question}

Predicted answer: {pred_answer}

Ground truth answer: {ground_truth}

Rules for judgment:
1. For numerical questions, any result within 3% of the ground truth answer is considered correct. Please compare abs(Predicted answer)/abs(True answer) with 3% to make your decision.
2. For multiple-choice questions, an exact match is required.
3. The answer should be clear and complete.
4. The calculation process alone is not considered correct.

Wrap your reasoning inside <thought></thought> and wrap the accuracy score inside <score></score> tags. Keep your reasoning concise, no more than 3-5 clear and informative sentences. Avoid repetition or unnecessary elaboration. Only output the reasoning and score using the required tags. Follow the output format as shown in the example below:

<thought>The predicted answer is 115624, which exactly matches the ground truth. The relative error is 0, well within the 3% threshold. The answer is clear, correct, and directly responds to the question.</thought>
<score>1</score>

Figure 13: Prompt for Reward Judge Model.

