# OpenReview forum: "Scaling Generalist Data-Analytic Agents"
_ICLR.cc/2026/Conference — ICLR 2026 Poster_

### Official Review · Reviewer_xdfR · 2025-10-27

**Soundness:** 3
**Presentation:** 2
**Contribution:** 3
**Rating:** 6
**Confidence:** 4

**Summary:**

This paper introduces DATAMIND, a framework for training open-source data-analytic agents. The paper tackles three major challenges:

- **Insufficient data resources:** proposes a fine-grained task taxonomy and a recursive query synthesis pipeline to generate diverse, multi-turn analytic problems, coupled with a knowledge-augmented, self-consistency-filtered data curation process that produces the DATAMIND-12K dataset.

- **Training strategy:** designs a hybrid SFT + RL objective with dynamic weighting to balance imitation and exploration during training.

- **Unstable multi-turn rollouts:** develops a memory-efficient rollout mechanism with asynchronous interaction and chunked code management to improve reliability in multi-turn, code-based roll-outs.

Fine-tuned with DATAMIND, a ReAct-style agent with Qwen-2.5-Coder-14B backbone achieves state-of-the-art results (average pass@1 = 71.16 across three benchmarks including DABench, TableBench, and BIRD), outperforming both open-source and proprietary baseline models including DeepSeek-V3.1 and GPT-5.

**Strengths:**

1. **Comprehensive framework:** the paper proposes a unified framework that integrates training data synthesis and SFT + RL fine-tuning for training open-source data-analytic agents, offering a reproducible recipe for scaling open-source data-analytic models.

2. **Strong empirical performance:** with DATAMIND, Qwen-2.5-Coder-14B model outperforms all baseline proprietary and open-source models, while Qwen-2.5-Coder-7B model achieves performance comparable to GPT-5.

3. **Comprehensive analyses:** the paper provides insightful ablation studies that offer practical understanding in curating high-quality training data and stabilizing the training procedure.

**Weaknesses:**

1. Since part of the training data is sourced from Kaggle, is there a possibility that the training data may overlap with public benchmarks used for evaluation? Some discussions or analysis to rule out such possibility would further enhance the validity of the results.

2. All expert trajectories are sampled from DeepSeek-V3.1, and GPT-4o-mini serves as the judge during both trajectory filtering and reward modeling. This raises concerns about introducing potential biases from these two models, and some discussions on how to mitigate such biases would be very helpful.

3. LLM-as-judge with GPT-4o-mini is used to evaluate the performance, although for some benchmarks objective metrics are available (e.g. Rouge-L for TableBench as described in Appendix D). This raises concerns about the reliability of the evaluation, especially when GPT-4o-mini is also used as the judge during training trajectory curation and reward modeling. Including objective metrics results would improve the robustness of the evaluation.

**Questions:**

1. For Rule-based Trajectory Filtering, especially 1) Format compliance and 3) Linguistic integrity, were these checks also performed using LLM-as-judge, or implemented through e.g. regular expression?

2. How is the format reward computed?

3. In table 1, Qwen-2.5 Coder-14B fine-tuned with TableLLM and Table-R1 perform worse on TableBench than the pretrained Qwen-2.5 Coder-14B (ReAct). Is there any insight on why fine-tuning on TableInstruct led to worse performance?

4. Regarding Fig 6, my understanding is that the conclusion “RL can narrow the performance gap between different base models” is based on models with the same backbone but different in SFT epochs (e.g., Qwen-7B with 1 v.s. 2 epochs). Could this conclusion extend to base models of different sizes (e.g., Qwen-7B v.s, Qwen-14B)?

---

> ### Author Response · Authors · 2025-11-19
> **Author Response (1/2)**
>
> Dear Reviewer dp72,
>
> We are deeply grateful for your valuable time and insightful feedback. Below are our detailed responses to your concerns:
>
> ### **W1: Data overlap.**
>
> To the best of our knowledge, the **DABench data files are harvested from GitHub**, whereas the **tables in TableBench are extracted from Wikipedia**, with neither overlapping with the corpora we curate. We sample the training split of BIRD and adopt synthetically generated schemas provided by OmniSQL. Consequently, **our sources are also disjoint from the BIRD test set**. To quantify this claim, we compute the **header-overlap ratio** between every table in DataMind-12K and the headers of each benchmark:
>
> |              | DABench | TableBench | BIRD  |
> | ------------ | ------- | ---------- | ----- |
> | **DataMind** | 1.29%   | 0.00%      | 0.02% |
>
> The resulting overlap is exactly 0% (or statistically indistinguishable from 0%) for every benchmark, demonstrating that **DataMind-12K introduces no data contamination into the evaluation**. In our revised manuscript, **we have added the analysis of this part in Appx. H.**
>
> ### **W2&3: Potential bias and the reliability of the judge model.**
>
> We adopt an LLM-as-judge paradigm because data-analysis outputs contain not only verifiable numbers but also long-form, descriptive explanations. **Traditional metrics such as Rouge-L are sensitive to surface lexical or sentence-level form and spurious tokens, yielding unreliable scores for open-ended answers**. Although some benchmarks impose rigid output templates that could simplify exact match evaluation, **we aim to train a general-purpose data-analysis agent rather than over-fit to any particular benchmark**; hence we keep the answer format open and let an LLM judge quality.
>
> To minimise judge bias, we carefully engineer the prompt (Fig. 13) with explicit instructions, fine-grained rubrics, and illustrative examples. The same judge model is used for every baseline to ensure fairness. During trajectory synthesis we inject analogous constraints into the agent instructions (Fig. 11), and we further penalise excessive length in the reward function (Eqn. 8) to discourage the agent from hacking the judge with verbose or hallucinated tokens.
>
> To further verify objectivity, we re-evaluate by replacing GPT-4o-mini with Qwen-2.5-72B. The results are shown below:
>
> |               |            | DABench |        | TableBench |        |
> | ------------- | ---------- | ------- | ------ | ---------- | ------ |
> | **Backbone**  | **Method** | pass@1  | pass@3 | pass@1     | pass@3 |
> | DeepSeek-V3.1 | ReAct      | 81.32   | 89.49  | 72.52      | 81.68  |
> | Qwen-2.5-72B  | ReAct      | 75.33   | 86.38  | 65.44      | 76.21  |
> | Qwen-2.5-7B   | DataMind   | 77.30   | 87.94  | 67.60      | 79.39  |
> | Qwen-2.5-14B  | DataMind   | 80.29   | 88.72  | 70.95      | 81.81  |
>
> We have incorporated these experiments into the main text (**Sec. 5.3, third paragraph**) and augmented the discussion with an analysis of **Fig. 5**. As shown, **the rankings produced by the two judges are virtually identical** (each model–dataset pair contributes two points, corresponding to pass@1 and pass@3). Quantitatively, **the Pearson correlation between the two sets of scores is 0.96**, indicating an almost perfect linear relationship.
>
> Furthermore, we isolate the descriptive-generation tasks in TableBench and evaluate the outputs of DataMind and the baselines with the **dataset’s original objective metric, Rouge-L**:
>
> |               |              | TableBench |        |
> | ------------- | ------------ | ---------- | ------ |
> | **Backbone**  | **Method**   | Rouge-L    | pass@1 |
> | DeepSeek-V3.1 | ReAct        | 19.64      | 72.52  |
> | Qwen-2.5-72B  | ReAct        | 17.93      | 65.44  |
> | Qwen-2.5-7B   | TableLLM     | 11.33      | 9.81   |
> |               | Table-R1     | 13.86      | 25.49  |
> |               | **DataMind** | 23.41      | 76.47  |
>
> We have incorporated these experiments into Appx. G.4. The resulting rankings align closely with those produced by our model-as-judge. Nevertheless, **Rouge-L clearly fails to capture true model performance**. It over-emphasizes surface lexical and sentence overlap with the gold label rather than answer correctness. This renders the performance differences among models marginal. Even the powerful DeepSeek-V3.1 attains a very low Rouge-L score, underscoring the rationality and fairness of employing a judge model for evaluation.

---

> ### Author Response · Authors · 2025-11-19
> **Author Response (2/2)**
>
> ### **Q1: Rule-based Trajectory Filtering.**
>
> All rule-based trajectory filtering is implemented via logical checks or regular expressions.
>
> For format compliance, we simply verify that special tokens are balanced; for instance, the number of `<think>` must equal the number of `</think>` in a trajectory.
>
> For linguistic integrity, we use a regular expression to detect the presence of any non-English characters.
>
> ### **Q2: How is the format reward computed?**
>
> The format-reward rule is identical to the format-compliance check described earlier: if the trajectory’s special-token pairing is invalid, the format reward is 0; otherwise it is 1.
>
> ### **Q3: Why 14B TableLLM and Table-R1 perform worse than ReAct on TableBench?**
>
> Thank you for this insightful observation! We believe the gain arises because the **14B coder backbone already possesses non-trivial planning and code-generation capabilities**; framing data analysis as code-driven ReAct rollouts is therefore a **distributionally aligned extension** of its pre-existing skills. By contrast, TableLLM and Table-R1 **concatenate the entire table content into the prompt** and ask the model to reason over raw text. This forces the LLM to **parse and manipulate heavily-structured data in a modality that was scarcely present in its pre-training corpus**, inducing **severe distribution shift** and degrading performance despite continued fine-tuning.
>
> A second factor may be **TableInstruct (training data of TableLLM and Table-R1)’s poor quality**. Its annotations were produced by simple CoT, SCot, or PoT templates followed by majority voting, so the resulting labels can be noisy and sub-optimal for robust training.
>
> ### **Q4: Could the conclusion in Figure 6 extend to different model sizes?**
>
> We further try the experiments in Fig. 6 on the 14B model. The results are shown below:
>
> | Epochs  | Cold Start-7B | RL-7B | diff  | Cold Start-14B | RL-14B | diff |
> | ------- | ------------- | ----- | ----- | -------------- | ------ | ---- |
> | zero    | 29.17         | 45.00 | 15.83 | 42.91          | 49.17  | 6.26 |
> | 1 epoch | 46.25         | 52.92 | 6.67  | 48.75          | 53.33  | 4.58 |
> | 2 epoch | 49.58         | 54.58 | 5.00  | 52.50          | 56.25  | 3.75 |
> | 3 epoch | 51.66         | 55.83 | 4.17  | 55.42          | 57.92  | 2.50 |
>
> The 14B model exhibits the same regularity: RL consistently **shrinks the performance gap across different initial checkpoints**, yet **rarely reverses their relative ranking**. Holding the number of epochs fixed, the 7B and 14B trajectories preserve this ordering, reinforcing the conclusion that **the preponderance of task-specific knowledge is acquired during SFT**, while **RL’s principal role is to unlock latent capabilities rather than to propel the model beyond its inherent capacity frontier**. We have added the corresponding analysis in our revised submission (**Sec. 5.4, fourth para**) and **re-draw the Fig. 6 (now Fig. 8) to reflect the newest results**.
>
>
>
> Thank you again for your constructive suggestions!
>
> **Please let us know if you have any further questions. If you find that our response addresses some of your concerns, would you kindly consider raising your rating score for our paper? We greatly appreciate your consideration.**
>
> Best Regards,
>
> Authors of DataMind

---

> > ### Author Response · Authors · 2025-11-27
> > **Kindly Reminder**
> >
> > Dear reviewer xdfR,
> >
> > We highly appreciate the constructive comments and insightful suggestions you have offered for our work. As the deadline of our discussion period is approaching, in order for us to have sufficient time to address any additional questions you may have, we kindly encourage you to engage in the ongoing discussion and share any further insights or clarifications you may have.
> >
> > **If you find that our response addresses some of your concerns, would you kindly consider raising your rating score for our paper?**
> >
> > Thank you very much for your time and consideration. We look forward to hearing from you soon.
> >
> > Best Regards,
> >
> > Authors of DataMind

---

### Official Review · Reviewer_dp72 · 2025-10-29

**Soundness:** 3
**Presentation:** 4
**Contribution:** 3
**Rating:** 8
**Confidence:** 5

**Summary:**

The paper presents DATAMIND, a data synthesis and training pipeline for building generalist data-analytic agents that operate through code-based multi-turn ReAct loops over heterogeneous files (.csv, .xlsx, .sqlite). The recipe includes: an 18-category task taxonomy with recursive easy-to-hard composition for query generation; knowledge-augmented trajectory sampling with self-consistency filtering and rule-based checks; a dynamically weighted SFT+RL objective (DAPO) with void-turn masking; and a memory-frugal rollout framework with sandboxed execution.

**Strengths:**

End-to-end, scalable pipeline. The work integrates dataset harvesting, fine-grained taxonomy, compositional query generation, trajectory vetting, and stable multi-turn rollout, offering a complete path from raw files to trained agents suitable for community reuse.

Training design that addresses common failure modes. The dynamic SFT+RL weighting, void-turn masking, and asynchronous chunked execution directly target instability in code-based multi-turn RL and are supported by ablations on reward/entropy dynamics.

Strong empirical results with broad format coverage. Consistent gains across three benchmarks and multiple file modalities, including conversion stress tests (e.g., TableBench/BIRD to unified formats), suggest robustness beyond narrow table QA.

**Weaknesses:**

Evaluation and reward rely on model-as-judge. Using GPT-4o-mini for both training rewards and final evaluation introduces circularity and potential stylistic bias; limited exact-match or verifier-based checks reduce measurement rigor, especially for descriptive answers.


Potential data overlap and protocol discrepancies. Training on BIRD/OmniSQL-adjacent corpora and converting evaluation benchmarks to new formats risk contamination or distribution shifts; fairness of comparisons may be affected by enforcing a single ReAct prompt across heterogeneous baselines.


Scope and generality claims are ahead of evidence. Results are confined to reasoning-oriented analytics; visualization, forecasting, and end-to-end scientific workflows are excluded. Claims of outperforming top proprietary models would benefit from additional third-party evaluators and held-out, verifier-checkable tasks.

**Questions:**

Please specify total RL budget: number of episodes, average steps per episode, total generated tokens, and wall-clock/compute (GPU type, days). The paper mentions ∼350 RL steps; what exactly constitutes a “step” here?

For the dynamic weighting γ, provide the full schedule, per-step values, and an ablation where γ is learned or adapted from validation signals instead of cosine.

How do SFT and RL gradients interact on overlapping tokens when a trajectory passes the RL filter? Are environment-emitted tokens always masked in both losses, and is there any credit assignment to code vs. natural-language tokens?

---

> ### Author Response · Authors · 2025-11-19
> **Author Response (1/3)**
>
> Dear Reviewer dp72,
>
> We are deeply grateful for your valuable time and insightful feedback. Below are our detailed responses to your concerns:
>
> ### **W1: Evaluation and reward rely on model-as-judge.**
>
> We adopt an LLM-as-judge paradigm because data-analysis outputs contain not only verifiable numbers but also long-form, descriptive explanations. **Traditional metrics such as Rouge-L are sensitive to surface lexical or sentence-level form and spurious tokens, yielding unreliable scores for open-ended answers**. Although some benchmarks impose rigid output templates that could simplify exact match evaluation, **we aim to train a general-purpose data-analysis agent rather than over-fit to any particular benchmark**; hence, we keep the answer format open and let an LLM judge quality.
>
> To minimise judge bias, we carefully engineer the prompt (Fig. 13) with explicit instructions, fine-grained rubrics, and illustrative examples. The same judge model is used for every baseline to ensure fairness. During trajectory synthesis we inject analogous constraints into the agent instructions (Fig. 11), and we further penalise excessive length in the reward function (Eqn. 8) to discourage the agent from hacking the judge with verbose or hallucinated tokens.
>
> To further verify objectivity, we re-evaluate by replacing GPT-4o-mini with Qwen-2.5-72B. The results are shown below:
>
> |               |            | DABench |        | TableBench |        |
> | ------------- | ---------- | ------- | ------ | ---------- | ------ |
> | **Backbone**  | **Method** | pass@1  | pass@3 | pass@1     | pass@3 |
> | DeepSeek-V3.1 | ReAct      | 81.32   | 89.49  | 72.52      | 81.68  |
> | Qwen-2.5-72B  | ReAct      | 75.33   | 86.38  | 65.44      | 76.21  |
> | Qwen-2.5-7B   | DataMind   | 77.30   | 87.94  | 67.60      | 79.39  |
> | Qwen-2.5-14B  | DataMind   | 80.29   | 88.72  | 70.95      | 81.81  |
>
> We have incorporated these experiments into the main text (**Sec. 5.3, third paragraph**) and augmented the discussion with an analysis of **Fig. 5**. As shown, **the rankings produced by the two judges are virtually identical** (each model–dataset pair contributes two points, corresponding to pass@1 and pass@3). Quantitatively, **the Pearson correlation between the two sets of scores is 0.96**, indicating an almost perfect linear relationship.
>
> Furthermore, we isolate the descriptive-generation tasks in TableBench and evaluate the outputs of DataMind and the baselines with the **dataset’s original objective metric, Rouge-L**:
>
> |               |              | TableBench |        |
> | ------------- | ------------ | ---------- | ------ |
> | **Backbone**  | **Method**   | Rouge-L    | pass@1 |
> | DeepSeek-V3.1 | ReAct        | 19.64      | 72.52  |
> | Qwen-2.5-72B  | ReAct        | 17.93      | 65.44  |
> | Qwen-2.5-7B   | TableLLM     | 11.33      | 9.81   |
> |               | Table-R1     | 13.86      | 25.49  |
> |               | **DataMind** | 23.41      | 76.47  |
>
> We have incorporated these experiments into Appx. G.4. The resulting rankings align closely with those produced by our model-as-judge. Nevertheless, **Rouge-L clearly fails to capture true model performance**. It over-emphasizes surface lexical and sentence overlap with the gold label rather than answer correctness. This renders the performance differences among models marginal. Even the powerful DeepSeek-V3.1 attains a very low Rouge-L score, underscoring the rationality and fairness of employing a judge model for evaluation.

---

> ### Author Response · Authors · 2025-11-19
> **Author Response (2/3)**
>
> ### **W2: Potential data overlap and protocol discrepancies.**
>
> To the best of our knowledge, the **DABench data files are harvested from GitHub**, whereas the **tables in TableBench are extracted from Wikipedia**, with neither overlapping with the corpora we curate (Kaggle). We sample the training split of BIRD and adopt synthetically generated schemas provided by OmniSQL. Consequently, **our sources are also disjoint from the BIRD test set**. To quantify this claim, we compute the **table header-overlap ratio** between every table in DataMind-12K and the headers of each benchmark:
>
> |              | DABench | TableBench | BIRD  |
> | ------------ | ------- | ---------- | ----- |
> | **DataMind** | 1.29%   | 0.00%      | 0.02% |
>
> The resulting overlap is exactly 0% (or statistically indistinguishable from 0%) for every benchmark, demonstrating that **DataMind-12K introduces no data contamination into the evaluation**. In our revised manuscript, **we have added the analysis of this part in Appx. H.**
>
> Regarding baseline evaluation, **we strictly adhere to each method’s original input–output protocol**.
>
> - For **TableLLM** and **Table-R1**, which expect the table serialized as a string inside the prompt, we convert every benchmark table into the exact delimiter-separated format prescribed by their released code before feeding it to the model.
> - For **OmniSQL** and **SQL-R1**, which are trained in the classical Text2SQL setting, we automatically dump each benchmark’s data files into a corresponding `.sqlite` database and prompt the models to emit a complete SQL statement.
>
> No post-processing or prompt engineering beyond the authors’ public instructions is applied. Consequently, the comparison is **protocol-faithful** and contains no unfair advantage. **We provide additional details on baseline reproduction in Appx. E line 1061.**
>
> ### **W3: Scope and generality claims are ahead of evidence.**
>
> Thank you for your critical feedback. As we explicitly state in Limitations (Appx. B, line 944), our current task scope is restricted to reasoning-oriented analytics; this is indeed a limitation of the present work. Extending the benchmark and agent to cover a broader spectrum of data-science tasks, such as visualization**, **forecasting, and scientific discovery, is already a top-priority item on our future-research agenda.
>
> **In our response to W1, we further verified the robustness and fairness of our evaluation method** by replacing the judge model and introducing a third-party metric, Rouge-L.
>
> **To further demonstrate the generalizability of our approach**, we evaluate DataMind-7B and DataMind-14B on a more challenging held-out benchmark called **QRData** [1]. It comprises both multiple-choice, numerical, and descriptive tasks, and knowledge-intensive causal-reasoning questions. We compare against several strong baselines and the results are presented below:
>
> |               |              | QRData    |           |
> | ------------- | ------------ | --------- | --------- |
> | **Backbone**  | **Method**   | pass@1    | pass@3    |
> | DeepSeek-V3.1 | ReAct        | 60.75     | 75.67     |
> | Qwen-2.5-72B  | ReAct        | 60.50     | 72.75     |
> | Qwen-2.5-7B   | **DataMind** | **57.66** | **69.34** |
> | Qwen-2.5-14B  | **DataMind** | **62.04** | **77.62** |
>
> Our 14B model still achieves the best performance, benefiting from the comprehensive coverage of data-analysis types and domains in DataMind-12K, as well as the consistent performance gains brought by our stable agent training strategy. Our 7B model also delivers competitive results. However, constrained by its parameter count, it lacks the domain-specific knowledge required, which prevents it from achieving equally impressive performance. Due to space constraints, **we have included the corresponding experiments and analyses in Appx. G.3 for your review.**
>
> [1] Are LLMs Capable of Data-based Statistical and Causal Reasoning? Benchmarking Advanced Quantitative Reasoning with Data. ACL 2024

---

> ### Author Response · Authors · 2025-11-19
> **Author Response (3/3)**
>
> ### **Q1: Specify total RL budge.**
>
> We apologize for the missing details regarding the RL budget. In Sec. 5.1 we already list the key hyper-parameters, including the batch size and learning rate used in RL training; the remaining details are given in Appx. F.
>
> Specifically, the RL stage runs for **1 epoch** with a batch size of **32**. With **11,707** training instances, this yields ⌈11 707 / 32⌉ = **366** gradient steps, which is why we report “**≈ 350 RL steps**”.
>
> During training, the average trajectory length per rollout is **1,500 tokens**. Each instance is rolled out **4 times**, so the total number of generated tokens is 1,500 × 4 × 11 707 ≈ **70.2 M tokens**.
>
> All experiments fit on a single server with **8 × 80 GB A100 GPUs**; one complete RL run takes approximately **1.5–2 days**.
>
> **We have incorporated the above information into Sec. 5.1 and Appx. F for the revised submission.**
>
> ### **Q2: The full schedule of $\gamma$.**
>
> **You can refer to the full schedule and the per-step value of $\gamma$ in the third subfigure of Figure 4**. It has a peak value of 0.9 and will gradually reduce to 0.05 based on a cosine schedule.
>
> We agree that making $\gamma$ learnable from validation signals could be a more optimal strategy. However, $\gamma$ primarily serves as a stabilizer, and identifying which validation signal (e.g., policy entropy or rollout accuracy) best reflects training stability remains challenging. We will investigate this issue in future work. Thank you for the valuable suggestion.
>
> ### **Q3: SFT and RL gradients on tokens.**
>
> In our loss computation, the SFT loss is computed over the expert trajectory, whereas the RL loss is computed solely over the rollout trajectory produced by the policy model. Consequently, the two losses are calculated on disjoint token sets and their gradient back-propagations do not overlap on tokens.
>
> For both losses we mask out all the tokens emitted by the environment.
>
> We do not distinguish between code and natural-language tokens, because for data-analysis tasks, reasoning and code-generation capabilities are equally critical, so they are treated identically.
>
>
>
> Thank you again for your constructive suggestions!
>
> **Please let us know if you have any further questions. We are very happy to continue the discussion!**
>
> Best Regards,
>
> Authors of DataMind

---

### Official Review · Reviewer_H48n · 2025-10-30

**Soundness:** 3
**Presentation:** 2
**Contribution:** 2
**Rating:** 4
**Confidence:** 4

**Summary:**

This paper addresses three key challenges in developing data analysis agents:
(i) the scarcity of training data resources,
(ii) the instability of long-horizon agent training and the unclear trade-off between SFT and RL training stages, and
(iii) the complexity of memory management in parallel agentic rollouts and multi-turn code generation.

To tackle these challenges, the authors propose:
(i) constructing a specialized training corpus,
(ii) dynamically combining SFT and RL losses with an adaptive weighting scheme, and
(iii) introducing an asynchronous agent generation and code execution framework with a chunk-wise code maintenance strategy to reduce peak memory usage.

**Strengths:**

1. The idea of building open-source data-analytic agents is interesting and timely.
2. The proposed system demonstrates promising results across multiple benchmarks.

**Weaknesses:**

1. It is unclear whether the observed performance improvements primarily stem from the proposed training methodology or from the curated dataset. A clearer ablation isolating these factors would strengthen the claim.
2. The benefit of dynamically combining SFT and RL remains ambiguous. From the rightmost subfigure in Figure 4, the training process appears to continuously shift from SFT to RL, although the paper does not follow an “SFT-first-then-RL” paradigm. It is not evident whether this dynamic approach yields tangible advantages over the “SFT-first-then-RL” paradigm. An ablation study is needed.

**Questions:**

Given the results in Table 1, it seems there is a clear upper bound on the performance across all models on the multiple benchmarks listed in the table. Do the authors have any insights into why this could happen and how to further improve the performance?

---

> ### Author Response · Authors · 2025-11-19
> **Author Response (1/2)**
>
> Dear Reviewer H48n,
>
> We are deeply grateful for your valuable time and insightful feedback. Below are our detailed responses to your concerns:
>
> ### **W1&2: Ablation studies on our curated dataset and training method.**
>
> Thank you very much for your insightful feedback. We sincerely apologize for omitting these crucial details; we have addressed them in the revised manuscript and do further explanation as follows.
>
> Indeed, Figure 3 already reports the result of fine-tuning a 7B model on DataMind-12K alone. We need to say that most of the performance improvement is from our curated dataset, as SFT alone can lift the accuracy to 62.54%, accounting for the bulk of the total improvement. The remaining gap is closed by subsequent RL. We introduce RL not only to boost extra points, but also to enhance out-of-distribution generalization and robustness to diverse, noisy environment feedback.
>
> However, applying conventional RL to data-analysis agents revealed three critical pain points:
>
> 1. **No principled schedule for mixing SFT and RL updates**. An imbalance either catastrophically forgets reasoning skills or prolongs exploration.
> 2. **Massively parallel file I/O and code execution** routinely crash the training stack.
> 3. **Distributional drift** caused by external feedback and compounding errors across multi-turn rollouts destabilizes policy optimization.
>
> Motivated by these observations, the technique described in Sec. 5.3 are designed to yield a **stable, long-horizon RL training**.
>
> **As for our curated dataset, we further conduct an ablation study on different training data sizes** by sampling different data volumes from DataMind-12K, with results shown below:
>
> | 7B model       | Avg.   |        |
> | -------------- | ------ | ------ |
> | **Data Sizes** | pass@1 | pass@3 |
> | 2K             | 55.76  | 72.86  |
> | 4K             | 62.79  | 76.55  |
> | 8K             | 66.13  | 78.04  |
> | 12K            | 68.10  | 79.07  |
>
> We have incorporated these experiments into the main text (**Sec. 5.3, first paragraph**) and augmented the discussion with an analysis of **Fig. 3**. As the training data volume increases, **the model’s performance exhibits a clear scaling law, demonstrating the scalability of the DataMind data-synthesis pipeline**. Another intriguing observation is that the gap between pass@1 and pass@3 gradually narrows as the data size grows. This pattern seems to align with the conclusion in [1]: RL training tends to increase the likelihood of generating correct samples, yet contributes little to expanding the model’s coverage of solvable problems.
>
> [1] Does reinforcement learning really incentivize reasoning capacity in llms beyond the base model? NeurIPS 2025
>
> **As for our training strategy**, we have added the following ablation experiments:
>
> | 7B model       | Avg.      |           |
> | -------------- | --------- | --------- |
> | **Strategy**   | pass@1    | pass@3    |
> | SFT            | 62.54     | 73.74     |
> | zero-RL        | 58.03     | 71.72     |
> | SFT-then-RL    | 63.42     | 75.46     |
> | **SFT-and-RL** | **68.10** | **79.07** |
>
> We have incorporated these experiments into the main text (**Sec. 5.3, second paragraph**) and augmented the discussion with an analysis of **Tab. 2**. We can see that zero-RL performs markedly worse than SFT, while SFT-then-RL yields only marginal gains over pure SFT.
>
> What these numbers do not reveal is the training stability conferred by our hybrid objective, as **both zero-RL and SFT-then-RL can easily fall into collapse**. In open-ended code-generation tasks, especially data analysis, unconstrained exploration frequently encounters fatal environment exceptions, causing entire roll-outs to collapse. When these corrupted trajectories are fed to the RL objective, the resulting gradient noise quickly destabilises training. We therefore **keep the SFT signal alive throughout RL**. Even though the mixing coefficient $\gamma$ becomes very small in later stages, it is **never set to zero**, so the SFT signal continues to supply a stabilising supervised anchor that suppresses divergence.

---

> ### Author Response · Authors · 2025-11-19
> **Author Response (2/2)**
>
> ### **Q1: Upperbound performance in Table 1 and how to improve?**
>
> Thank you for your careful observations! **After systematically analyzing the outputs of various models and baselines on several benchmarks**, we guess that a latent performance ceiling may be shared by all existing systems, attributable to the following factors:
>
> 1. **Benchmark quality.** Both the construction of DABench and TableBench rely on answer annotations generated by LLMs; consequently, the labels unavoidably contain noise and outright errors that cap the achievable accuracy.
>
> 2. **Difficulty gap.** BIRD is curated by human experts and includes multi-table SQL statements of up to ~1K lines—queries that remain beyond the reach of even the strongest open-source models. Models trained on ordinary data struggle to bridge this complexity chasm.
>
> 3. **Commonsense deficit.** Several data files stem from highly specialized vertical domains whose analysis demands external commonsense knowledge in addition to reasoning and code-generation skills. Most LLMs lack this domain knowledge, further limiting performance.
>
> To address these challenges, we outline three directions and welcome your critique:
>
> 1. **Answer scaling.** Some rapid developing agentic fields (information-seeking, GUI, and software-engineering agents) is driven by the ease of obtaining ground-truth answers; synthetic data are typically constructed by starting from the answer and reverse-engineering plausible questions. Data-analysis agents face the opposite situation—questions are easy to generate, but correct answers are expensive. Developing scalable strategies to produce high-fidelity answers is therefore a prerequisite for data-centric agent learning.
>
> 2. **Dense reward design.** When problems become long-horizon and combinatorially complex, a sparse terminal reward is no longer informative. Designing general-purpose PRMs that provide fine-grained learning signals is an open and critical challenge .
>
> 3. **Agent-centric pre-training.** Agents must acquire domain-specific priors before they can effectively fine-tune or interact. Continual or masked pre-training objectives that bake in planning, tool-use, and multi-agent coordination biases have recently shown strong scaling properties and zero-shot generalization , suggesting that foundation models should be grown, not merely patched, with agentic capabilities.
>
>
>
> Thank you again for your constructive suggestions!
>
> **Please let us know if you have any further questions. If you find that our response addresses some of your concerns, would you kindly consider raising your rating score for our paper? We greatly appreciate your consideration.**
>
> Best Regards,
>
> Authors of DataMind

---

> ### Author Response · Authors · 2025-11-26
> **Kindly Reminder**
>
> Dear reviewer H48n,
>
> We highly appreciate the constructive comments and insightful suggestions you have offered for our work. As the discussion period has only a few days left, in order for us to have sufficient time to address any additional questions you may have, we kindly encourage you to engage in the ongoing discussion and share any further insights or clarifications you may have.
>
> **If you find that our response addresses some of your concerns, would you kindly consider raising your rating score for our paper?**
>
> Thank you very much for your time and consideration. We look forward to hearing from you soon.
>
> Best Regards,
>
> Authors of DataMind

---

### Official Review · Reviewer_dfWg · 2025-10-31

**Soundness:** 3
**Presentation:** 2
**Contribution:** 3
**Rating:** 6
**Confidence:** 2

**Summary:**

This paper proposes a pipeline for constructing generalist data-science agents, addressing key challenges such as data scarcity, unstable multi-turn rollout, and balancing RL and SFT training. The authors curate a 12K-sample dataset and train models on it. The resulting DATAMIND-7B/14B models show promising performance on several data-analytic benchmarks.

**Strengths:**

The paper is well-motivated, and the proposed pipeline is comprehensive, combining a thoughtful data-synthesis process with a robust training strategy. The experimental results are promissing and the authors also analyze useful empirical insights into agent training.

**Weaknesses:**

1. Missing baseline comparisons: The paper does not include comparisons with other data-analysis agents such as DS-Agent or Data-Interpreter, which would strengthen the empirical claims.
2. Limited base model and benchmark diversity: The model is trained only on Qwen, which may introduce family bias and inflate the reported performance. Moreover, evaluation is restricted to three tabular/SQL-focused benchmarks; the generalization to broader data-analysis scenarios remains unclear.
3. Additional studies on different base models, dataset sizes, and the impact of recursive composition depth would make the analysis more comprehensive.

**Questions:**

1. Since gold answers are already available, why use a judge model for evaluation? Could this introduce noise, and how might that be mitigated?
2. Can DATAMIND handle multimodal inputs (e.g., CSV + plots), or is it limited to text-based tabular data?

---

> ### Author Response · Authors · 2025-11-19
> **Author Response (1/3)**
>
> Dear Reviewer dfWg,
>
> We are deeply grateful for your valuable time and insightful feedback. Below are our detailed responses to your concerns:
>
> ### **W1: Missing baseline comparisons.**
>
> We think DS-Agent and Data-Interpreter are **prompt- and scaffold-level refinements tailored to powerful closed-source models**, whereas our work pursues the opposite goal: training small open-source LMs to acquire general-purpose data-analysis skills with the simplest possible ReAct pipeline. Consequently, **the two lines of research are orthogonal rather than directly comparable**.
>
> Nonetheless, we agree that additional baselines can further strengthen our paper. After carefully reviewing DS-Agent, we found that its gains hinge on **task-specific code templates engineered for its own benchmark** and **cannot transfer to other datasets**. We therefore omit it and instead supplement experiments with Data Interpreter.
>
> **Data Interpreter** has been embedded into the MetaGPT framework [1] as an off-the-shelf analytics tool. Despite our best effort to decouple its pipeline and run it in isolation, we are unable to reproduce the reported 94.93% GPT-4o score on DABench and **it also cannot be adapted to SQL tasks**. The same discrepancy has been independently acknowledged in another paper [2]. Our reproduction results are as detailed below:
>
> |               |                  | DABench |        | TableBench |        |
> | ------------- | ---------------- | ------- | ------ | ---------- | ------ |
> | **Backbone**  | **Method**       | pass@1  | pass@3 | pass@1     | pass@3 |
> | DeepSeek-V3.1 | ReAct            | 81.32   | 89.49  | 72.52      | 81.68  |
> |               | Data-Interpreter | 67.70   | 80.93  | 61.98      | 74.52  |
> | Qwen-2.5-72B  | ReAct            | 75.33   | 86.38  | 65.44      | 76.21  |
> |               | Data-Interpreter | 62.26   | 75.10  | 49.90      | 58.84  |
> | Qwen-2.5-7B   | ReAct            | 15.05   | 35.41  | 11.70      | 28.63  |
> |               | Data-Interpreter | 54.09   | 65.37  | 27.99      | 41.82  |
> |               | DataMind         | 77.30   | 87.94  | 67.60      | 79.39  |
> | Qwen-2.5-14B  | ReAct            | 71.21   | 83.27  | 56.96      | 69.97  |
> |               | Data-Interpreter | 60.70   | 74.71  | 40.71      | 55.82  |
> |               | DataMind         | 80.29   | 88.72  | 70.95      | 81.81  |
>
> It can be found that task-specific scaffolds do not consistently outperform the vanilla ReAct paradigm, and prompts engineering for one model may fail to generalize to others. We believe that the greatest truths are the simplest. This is also why we believe **agent training**, rather than prompt engineering and scaffold constructing, is the more promising path. **We have included the corresponding experiments and analyses in Appx. G.2 for your review.**
>
> [1] https://github.com/FoundationAgents/MetaGPT.
>
> [2] DatawiseAgent: A Notebook-Centric LLM Agent Framework for Adaptive and Robust Data Science Automation. EMNLP 2025
>
> ### **W2: Limited base model and benchmark diversity**
>
> We primarily conduct experiments with the Qwen family, as it is widely acknowledged that Qwen exhibits stronger reasoning capacity and higher plasticity during RL training [3, 4]. **To verify the universality of our DataMind pipeline on different kinds of base models**, we additionally train **Llama-3.1-8B-Instruct** on DataMind-12K and the results are reported below:
>
> |                  |              | DABench |        | TableBench |        | BIRD   |        | Avg    |        |
> | ---------------- | ------------ | ------- | ------ | ---------- | ------ | ------ | ------ | ------ | ------ |
> | **Backbone**     | **Method**   | pass@1  | pass@3 | pass@1     | pass@3 | pass@1 | pass@3 | pass@1 | pass@3 |
> | DeepSeek-V3.1    | ReAct        | 81.32   | 89.49  | 72.52      | 81.68  | 57.89  | 68.12  | 70.58  | 79.76  |
> | GPT-5            | ReAct        | 78.21   | 85.21  | 69.93      | 78.37  | 60.17  | 65.19  | 69.44  | 76.26  |
> | Llama-3.3-70B    | ReAct        | 69.78   | 80.16  | 55.47      | 70.36  | 59.10  | 68.58  | 61.45  | 73.03  |
> | Qwen-2.5-72B     | ReAct        | 75.33   | 86.38  | 65.44      | 76.21  | 60.30  | 69.49  | 67.02  | 77.36  |
> | Qwen-2.5-7B      | DataMind     | 77.30   | 87.94  | 67.60      | 79.39  | 59.41  | 69.88  | 68.10  | 79.07  |
> | **Llama-3.1-8B** | **DataMind** | 73.67   | 87.16  | 64.49      | 76.46  | 58.52  | 69.56  | 65.56  | 77.73  |
>
> It can be observed that the DataMind pipeline transfers robustly across model families. Llama-8B not only surpasses Llama-70B but also performs on par with strong baselines. Its pass@3 further approaches the level of DeepSeek-V3.1 and DataMind-7B. Due to space constraints, **we have included the corresponding experiments and analyses in Appx. G.1 for your review.**
>
> [3] Cognitive Behaviors that Enable Self-Improving Reasoners, or, Four Habits of Highly Effective STaRs. COLM 2025
>
> [4] OctoThinker: Mid-training Incentivizes Reinforcement Learning Scaling. arXiv2025

---

> ### Author Response · Authors · 2025-11-19
> **Author Response (2/3)**
>
> **To further demonstrate the generalizability of our approach**, we evaluate DataMind-7B and DataMind-14B on a more challenging benchmark called **QRData** [5]. It comprises both multiple-choice, numerical, and descriptive tasks, and knowledge-intensive causal-reasoning questions. We compare against several strong baselines and the results are presented below:
>
> |               |              | QRData    |           |
> | ------------- | ------------ | --------- | --------- |
> | **Backbone**  | **Method**   | pass@1    | pass@3    |
> | DeepSeek-V3.1 | ReAct        | 60.75     | 75.67     |
> | Qwen-2.5-72B  | ReAct        | 60.50     | 72.75     |
> | Qwen-2.5-7B   | **DataMind** | **57.66** | **69.34** |
> | Qwen-2.5-14B  | **DataMind** | **62.04** | **77.62** |
>
> Our 14B model still achieves the best performance, benefiting from the comprehensive coverage of data-analysis types and domains in DataMind-12K, as well as the consistent performance gains brought by our stable agent training strategy. Our 7B model also delivers competitive results. However, constrained by its parameter count, it lacks the domain-specific knowledge required, which prevents it from achieving equally impressive performance. Due to space constraints, **we have included the corresponding experiments and analyses in Appx. G.3 for your review.**
>
> [5] Are LLMs Capable of Data-based Statistical and Causal Reasoning? Benchmarking Advanced Quantitative Reasoning with Data. ACL 2024
>
> ### **W3: Additional studies on different base models, dataset sizes, and recursive composition depth**
>
> Thank you for the valuable suggestion! In W2 we have added experiments with Llama-3.1-8B-Instruct to analyze the effect of varying the base model. We further conduct **an ablation study on different training data sizes** by sampling different data volumes from DataMind-12K, with results shown below:
>
> | 7B model       | Avg.   |        |
> | -------------- | ------ | ------ |
> | **Data Sizes** | pass@1 | pass@3 |
> | 2K             | 55.76  | 72.86  |
> | 4K             | 62.79  | 76.55  |
> | 8K             | 66.13  | 78.04  |
> | 12K            | 68.10  | 79.07  |
>
> We have incorporated these experiments into the main text (**Sec. 5.3, first paragraph**) and augmented the discussion with an analysis of **Fig. 3**. As the training data volume increases, **the model’s performance exhibits a clear scaling law, demonstrating the scalability of the DataMind data-synthesis pipeline**. Another intriguing observation is that the gap between pass@1 and pass@3 gradually narrows as the data size grows. This pattern seems to align with the conclusion in [6]: RL training tends to increase the likelihood of generating correct samples, yet contributes little to expanding the model’s coverage of solvable problems.
>
> [6] Does reinforcement learning really incentivize reasoning capacity in llms beyond the base model? NeurIPS 2025
>
> Our data-volume ablation already demonstrates that the proposed data synthesis pipeline can effectively scale: once a sufficient volume of “easy” instances is reached, additional simple examples yield negligible gains, and further improvement can only come from harder, recursively composed data. **This result indirectly corroborates the effectiveness of our recursive composition.**

---

> ### Author Response · Authors · 2025-11-19
> **Author Response (3/3)**
>
> ### **Q1. Why use a judge model for evaluation?**
>
> We adopt an **LLM-as-judge** protocol because data-analysis outputs contain not only verifiable numbers but also long-form, descriptive explanations. **Traditional metrics such as Rouge-L are highly sensitive to surface form and spurious tokens**, yielding unreliable scores for open-ended answers. Although some benchmarks impose rigid output templates that facilitate exact matching, **our goal is to train a general-purpose data-analysis agent rather than over-fit to any single benchmark format; therefore we allow free-form answers and use a strong language model as the evaluator**.
>
> To minimise judge bias, we carefully engineer the prompt (Figure 13): it contains explicit instructions, fine-graded rules, and illustrative output examples. The **same prompt and judge model are used for every baseline to ensure fairness**. To further verify objectivity, we re-evaluate by replacing GPT-4o-mini with Qwen-2.5-72B. The results are shown below:
>
> |               |            | DABench |        | TableBench |        |
> | ------------- | ---------- | ------- | ------ | ---------- | ------ |
> | **Backbone**  | **Method** | pass@1  | pass@3 | pass@1     | pass@3 |
> | DeepSeek-V3.1 | ReAct      | 81.32   | 89.49  | 72.52      | 81.68  |
> | Qwen-2.5-72B  | ReAct      | 75.33   | 86.38  | 65.44      | 76.21  |
> | Qwen-2.5-7B   | DataMind   | 77.30   | 87.94  | 67.60      | 79.39  |
> | Qwen-2.5-14B  | DataMind   | 80.29   | 88.72  | 70.95      | 81.81  |
>
> We have incorporated these experiments into the main text (**Sec. 5.3, third paragraph**) and augmented the discussion with an analysis of **Fig. 5**. As shown, **the rankings produced by the two judges are virtually identical** (each model–dataset pair contributes two points, corresponding to pass@1 and pass@3). Quantitatively, **the Pearson correlation between the two sets of scores is 0.96**, indicating an almost perfect linear relationship.
>
> Furthermore, we isolate the descriptive-generation tasks in TableBench and evaluate the outputs of DataMind and the baselines with the dataset’s original objective metric, **Rouge-L**:
>
> |               |              | TableBench |        |
> | ------------- | ------------ | ---------- | ------ |
> | **Backbone**  | **Method**   | Rouge-L    | pass@1 |
> | DeepSeek-V3.1 | ReAct        | 19.64      | 72.52  |
> | Qwen-2.5-72B  | ReAct        | 17.93      | 65.44  |
> | Qwen-2.5-7B   | TableLLM     | 11.33      | 9.81   |
> |               | Table-R1     | 13.86      | 25.49  |
> |               | **DataMind** | 23.41      | 76.47  |
>
> We have incorporated these experiments into Appx. G.4. The resulting rankings align closely with those produced by our model-as-judge. Nevertheless, **Rouge-L clearly fails to capture true model performance**. It over-emphasizes surface lexical and sentence overlap with the gold label rather than answer correctness. This renders the performance differences among models marginal. Even the powerful DeepSeek-V3.1 attains a very low Rouge-L score, underscoring the rationality and fairness of employing a judge model for evaluation.
>
> ### **Q2. Can DataMind handle multimodal inputs?**
>
> We sincerely apologize that the current version of DataMind only accepts tabular data files and textual questions. Your suggestion is an excellent future direction. We have added this limitation to the revised manuscript (**Appx. B, line 971**) and will extend DataMind to additional modalities in our subsequent work.
>
>
>
> Thank you again for your constructive suggestions!
>
> **Please let us know if you have any further questions. If you find that our response addresses some of your concerns, would you kindly consider raising your rating score for our paper? We greatly appreciate your consideration.**
>
> Best Regards,
>
> Authors of DataMind

---

> > ### Author Response · Authors · 2025-11-27
> > **Kindly Reminder**
> >
> > Dear reviewer dfWg,
> >
> > We highly appreciate the constructive comments and insightful suggestions you have offered for our work. As the deadline of our discussion period is approaching, in order for us to have sufficient time to address any additional questions you may have, we kindly encourage you to engage in the ongoing discussion and share any further insights or clarifications you may have.
> >
> > **If you find that our response addresses some of your concerns, would you kindly consider raising your rating score for our paper?**
> >
> > Thank you very much for your time and consideration. We look forward to hearing from you soon.
> >
> > Best Regards,
> >
> > Authors of DataMind

---

### Author Response · Authors · 2025-11-19
**Global Response (2/2)**

**For Reviewer dp72:**

- Add the ablation on evaluation method (change the judge model/metrics to Qwen-72B and Rouge-L). (Page 8, Sec. 5.3, Para 3 and Fig. 5; Page 22-23, Appx. G.4 and Tab. 7)
- Add the analysis about potential data overlap. (Page 23, Appx. H and Tab. 8)
- Add more detailed baseline reproduction. (Page 20, Appx. E, Line 1074)
- Add the results of DataMind on QRData benchmark. (Page 22, Appx. G.3 and Tab. 6)
- Add more detailed RL budget. (Page 7, Sec. 5.1, Para 3; Page 21, Appx. F and Tab. 3)

**For Reviewer xdfR:**

- Add the analysis about potential data overlap. (Page 23, Appx. H and Tab. 8)
- Add the ablation on evaluation method (change the judge model/metrics to Qwen-72B and Rouge-L). (Page 8, Sec. 5.3, Para 3 and Fig. 5; Page 22-23, Appx. G.4 and Tab. 7)
- Add the 14B model experiments in the previous Fig. 6 (now Fig. 8). (Page 9, Sec. 5.3, Para 4 and Fig. 8)

**We will continue to further enhance the quality of this paper according to the discussion with all the reviewers.**

# Our motivation and contributions

Last but not least, we wish to **reiterate the motivation and main contributions** of our paper.

Current data-analytic agents rely heavily on prompt engineering over proprietary models, while open-source models struggle to face diverse-format, large-scale data files and long-horizon, multi-step reasoning that real-world analytics demands.

This paper introduces DataMind, a scalable data synthesis and agent training recipe designed to build generalist data-analytic agents. **DataMind tackles three key challenges** in building open-source data-analytic agents, including *insufficient data resources*, *improper training strategy*, and *unstable code-based multi-turn rollout*.

**Concretely, DataMind applies:**

1. A fine-grained task taxonomy and a recursive easy-to-hard task composition mechanism to increase the diversity and difficulty of synthesized queries;
2. A knowledge-augmented trajectory sampling strategy followed by model-based and rule-based filtering;
3. A dynamically adjustable training objective combining both SFT and RL losses;
4. A memory-frugal and stable code-based multi-turn rollout framework.

Built on DataMind, we curate **DataMind-12K**, a high-quality trajectory set spanning diverse domains, task categories, and data file formats for data-analytic tasks. Trained on DataMind-12K, our **DataMind-14B** achieves state-of-the-art with an average score of 71.16% on multiple data analysis benchmarks, outperforming the strongest proprietary baselines DeepSeek-V3.1 and GPT-5. Our **DataMind-7B** also performs best among all open-source models with a score of 68.10%. **We also incorporate some empirical insights gained from our exploratory trials into the analysis experiments, aiming to provide actionable insights about agentic training for the community. We will release DataMind-12K and DataMind-7B,14B for the community's future research.**

# Thank you!

We sincerely hope our responses and revisions address all reviewers’ concerns. We sincerely believe that these updates may help us better deliver the benefits of the proposed work to the ICLR community. In conclusion, we extend our heartfelt thanks to all the reviewers for their invaluable suggestions, which have significantly enhanced the rigor and completeness of our work. We welcome further discussion and are more than willing to answer any additional questions you may have. Thank you very much!

---

### Author Response · Authors · 2025-11-19
**Global Response (1/2)**

Dear all reviewers,

Thank you for your thoughtful reviews!

# Our Strengths Summarized by Reviewers

First, we appreciate all of your **positive comments** highlighting the strengths of our work for a summary:

- **Well-motivated, timely idea**
  - "The paper is well-motivated" (Reviewer dfWg)
  - "The idea of building open-source data-analytic agents is interesting and timely." (Reviewer H48n)
- **Comprehensive framework**
  - "The proposed pipeline is comprehensive, combining a thoughtful data-synthesis process with a robust training strategy" (Reviewer dfWg)
  - "End-to-end, scalable pipeline, offering a complete path from raw files to trained agents suitable for community reuse." (Reviewer dp72)
  - "Comprehensive framework, offering a reproducible recipe for scaling open-source data-analytic models." (Reviewer xdfR)
- **Promising results**
  - "The experimental results are promissing" (Reviewer dfWg)
  - "The proposed system demonstrates promising results across multiple benchmarks." (Reviewer H48n)
  - "Strong empirical results with broad format coverage." (Reviewer dp72)
  - "Strong empirical performance" (Reviewer xdfR)
- **Insightful, useful analysis**
  - " The authors also analyze useful empirical insights into agent training." (Reviewer dfWg)
  - "Comprehensive analyses: the paper provides insightful ablation studies" (Reviewer xdfR)

# Our main revisions and supplementary experiments

We also sincerely thank reviewers for your constructive feedback and questions to improve our manuscript. In response to the reviewers' concerns and questions, **we mainly conduct the following revisions (A revised draft of our manuscript has been uploaded, with added content marked in blue)**:

**For Reviewer dfWg:**

- Add the results of Data Interpreter. (Page 22, Appx. G.2 and Tab. 5)
- Add the results of DataMind on Llama-3.1-8B-Instruct model. (Page 21-22, Appx. G.1 and Tab. 4)
- Add the results of DataMind on QRData benchmark. (Page 22, Appx. G.3 and Tab. 6)
- Add the ablation study on data sizes. (Page 7-8, Sec. 5.3, Para 1 and Fig. 3)
- Add the ablation on evaluation method (change the judge model/metrics to Qwen-72B and Rouge-L). (Page 8, Sec. 5.3, Para 3 and Fig. 5; Page 22-23, Appx. G.4 and Tab. 7)
- Add a limitation about multimodal data formats. (Page 18, Appx. B, Line 971)

**For Reviewer H48n:**

- Add the ablation study on data sizes. (Page 7-8, Sec. 5.3, Para 1 and Fig. 3)
- Add the ablation study on training strategy. (Page 8, Sec. 5.3, Para 2 and Tab. 2)

---

### Author Response · Authors · 2025-12-01
**Summary of Our Rebuttal**

Dear Area Chair,

We would like to express our sincere gratitude for the time you have devoted to our submission under these exceptional circumstances. To help minimize your workload, we respectfully provide a concise summary of our rebuttal:

## **Regarding our rating scores**

Our initial ratings are 8-6-6-4, yielding a mean of 6.0. According to a fully-legal, fully-anonymous, and widely-used score statistical [website](https://papercopilot.com/statistics/iclr-statistics/iclr-2026-statistics/), **our paper lies in the top 3.84%–9.21% of all submissions, well above the conference’s approximate 30% acceptance threshold**.

The only negative review (Reviewer H48n, rating 4) raised two concerns: (i) the absence of ablations on data and (ii) the lack of ablations on training strategies. We have conducted both sets of supplementary experiments, included the corresponding plots and statistical analyses, and integrated these results into the revised manuscript. Reviewer H48n also posed an open question on how performance might be further improved. Drawing on our empirical observations, we offer three concrete causes and three actionable future directions. **We believe these additions have fully addressed the reviewer’s concerns.**

We regret that, **despite our active engagement, none of the reviewers replied during the whole discussion phase**. Consequently, we respectfully summarize our rebuttal for your convenience.

## **Regarding reviewers' concerns and our responses**

**Please note that the expanded, point-by-point rebuttal can be found in the Global Response that begins in the next window and the Author Response below each reviewer's reviews.** We respectfully invite you to consult them at your convenience.

Global concerns

1. *LLM-as-judge.* Three of the four reviewers expressed their concerns about our use of an LLM judge during both training and evaluation. To demonstrate our LLM judge's rationality, we adopted two complementary experiments:

   1) We employed different base models to act as judges at train and test time, thereby avoiding any “shared-brain” bias.

   2) We cross-checked the results with the more traditional Rouge-L metric.

   All conclusions remain consistent under this stricter protocol, and a formal correlation analysis further confirms the alignment between all the two assessment schemes.

2. *Potential data overlap.* Two of the four reviewers wondered whether the data files we collected from Kaggle, BIRD and OmniSQL might accidentally overlap with thetest sets. We carefully traced the provenance of every test query and table, demonstrating that our training files and the official benchmarks draw from disjoint sources. In addition, we ran a large-scale statistical similarity check on table schemas and contents; the resulting overlap is statistically indistinguishable from zero.

Beyond the global issues, we have responded to each reviewer’s individual suggestions as follows.

- Reviewer dfWg (rating 6, confidence 2)
  The reviewer encouraged us to include additional baselines, models and benchmarks. We have run the requested experiments and integrated the corresponding analyses into both the rebuttal and the revised paper.

- Reviewer H48n (rating 4, confidence 4)
  The reviewer asked for ablations on our data composition and training strategy. All suggested ablative studies have been completed and are reported in detail in the rebuttal and in the updated manuscript.

- Reviewer dp72 (rating 8, confidence 5)
  The reviewer suggested evaluating on held-out and verifiable tasks. We therefore texted on QRData, a new test set, and re-assessed our approach with Rouge-L as an objective metric on the original tasks as well. Minor hyper-parameter and methodological clarifications requested by the reviewer are also addressed point-by-point.

- Reviewer xdfR (rating 6, confidence 4)
  The reviewer raised several implementation details and questioned whether the conclusion of our final analysis still holds for 14B model. We have supplied extra experiments, refined the exposition, and updated the paper accordingly.

**We respectfully invite you to examine the full rebuttal (global response plus individual replies) and the revised paper**, in which all changes are highlighted in blue for easy reference. **Given our strong initial scores and the additional evidence provided, we sincerely hope you will find our paper worthy of acceptance.**

Thank you very much again for your invaluable time and consideration!

Best Regards,

Authors of DataMind

---

### Meta-Review · Area_Chair_Jisj · 2025-12-30

**Summary:**

This paper introduces DATAMIND, a synthesis and training pipeline that can be used to build generalist data-analytic agents. This paper addressed three key challenges in the development of such agents. First the scarcity of data resources, the instability of long horizon training, and the complexity of memory management in parallel agentic rollouts. The reviewers broadly agreed that this work is well motivated, and the contributions meet the bar for presentation at ICLR. During the rebuttal the authors compared with baselines addressing reviewer’s concerns. This work meets the bar of presentation at ICLR.

**Reviewer Concerns:**

The reviewers raised concerns about missing baselines. Most of these concerns have been addressed in the rebuttal.

**Reviewer Scores:**

The reviewer's scores were already at an accept level at the start of the discussion.

---

### Decision · Program_Chairs · 2026-01-26

Accept (Poster)